# Harnessing virus flexibility to selectively capture and profile rare circulating target cells for precise cancer subtyping

Hui-Da Li[1], Yuan-Qiang Chen[2], Yan Li[3], Xing Wei[1], Si-Yi Wang[1], Ying Cao[1], Rui Wang[1], Cong Wang[4], Jing-Yue Li[4], Jian-Yi Li[4] ✉, Hong-Ming Ding [2] ✉, Ting Yang [1] ✉, Jian-Hua Wang[1] & Chuanbin Mao [5] ✉

The effective isolation of rare target cells, such as circulating tumor cells, from whole blood is still challenging due to the lack of a capturing surface with strong target-binding affinity and non-target-cell resistance. Here we present a solution leveraging the flexibility of bacterial virus (phage) nanofibers with their sidewalls displaying target circulating tumor cell-specific aptamers and their ends tethered to magnetic beads. Such flexible phages, with low stiffness and Young's modulus, can twist and adapt to recognize the cell receptors, energetically enhancing target cell capturing and entropically discouraging non-target cells (white blood cells) adsorption. The magnetic beads with flexible phages can isolate and count target cells with significant increase in cell affinity and reduction in non-target cell absorption compared to magnetic beads having rigid phages. This differentiates breast cancer patients and healthy donors, with impressive area under the curve (0.991) at the optimal detection threshold (>4 target cells mL$^{-1}$). Immunostaining of captured circulating tumor cells precisely determines breast cancer subtypes with a diagnostic accuracy of 91.07%. Our study reveals the power of viral mechanical attributes in designing surfaces with superior target binding and non-target anti-fouling.

Affinity-based surface bioassays, such as enzyme-linked immunosorbent assays (ELISAs) and immunomagnetic isolation, hold crucial significance in clinical diagnostics, environmental monitoring, and drug screening. The initial phase of these assays hinges on the interaction between ligands and receptors, dictating the assay's affinity. Considerable efforts have been directed towards augmenting binding affinity by tailoring parameters such as ligand type, density, distribution, and conformation[1–3]. Particularly noteworthy is the challenge posed by complex biological samples containing various matrix components (e.g., blood), where non-specific adsorption of non-target cells or biomolecules significantly undermines assay performance. This occurs either by occupying active target-binding sites or obstructing on-surface signal transduction. Consequently, a pressing need has emerged for the advancement of surface-shielding strategies[4–6]. The application of surface coatings composed of anti-fouling polymers like poly(ethylene glycol) (PEG) or zwitterionic peptides has demonstrated efficacy in reducing non-specific adsorption[7–9]. However, the simultaneous presence of anti-fouling polymers and

[1]Research Center for Analytical Sciences, Department of Chemistry, College of Sciences, Northeastern University, Box 332, Shenyang 110819, China. [2]Center for Soft Condensed Matter Physics and Interdisciplinary Research, Soochow University, Suzhou 215006, China. [3]Department of Periodontology, The Second Affiliated Hospital, College of Medicine, Zhejiang University, Hangzhou 310009, China. [4]Department of Breast Surgery, Liaoning Cancer Hospital & Institute, Cancer Hospital of China Medical University, Shenyang 110042, China. [5]Department of Biomedical Engineering, The Chinese University of Hong Kong, Shatin, Hong Kong SAR, China. ✉e-mail: sjbreast@yeah.net; dinghm@suda.edu.cn; yangting@mail.neu.edu.cn; cmao@cuhk.edu.hk

affinity ligands on surfaces unavoidably leads to mutual interference, potentially compromising target-binding affinity or anti-fouling capabilities[10–12].

M13 bacteriophage (also known as phage) presents itself as a nanofiber-like, bacteria-specific virus, measuring 880 nm in length and 6 nm in diameter. Its single-stranded DNA resides within a coat comprised of five distinct capsid proteins. Among these, a major capsid protein (pVIII, approximately 2700 copies) is helically arranged along the length, while four minor capsid proteins (5 copies each) cap both ends, with two of these proteins situated at a single tip[13,14]. Notably, the surface of M13 nanofibers can be easily tailored through genetic engineering or chemical modification. This versatility enables the orthogonal functionalization of different capsid proteins, yielding desired functionalities—a unique attribute when compared to other artificial analogs. Consequently, the assembly of M13 into scaffolds has yielded remarkable successes in diverse applications[15], including tissue regeneration[16,17], therapy[18–20], sensing[21,22], and even energy harvesting[23]. We recently discovered that M13 phage is capable of accelerating the mass transport at the liquid-solid interface due to its sway motion[24]. This encourages us to further explore its mechanical properties. Despite the predominant focus on the biochemical traits of M13 phage in most studies, scant attention has been directed towards the potential impact of its physical or mechanical properties on cellular interactions. This gap in understanding is particularly relevant to endeavors involving the isolation of circulating tumor cells (CTCs).

CTCs, originating from solid tumors and entering the bloodstream, hold significant promise as targets for liquid biopsy in diagnosing tumor metastases and assessing prognosis[25,26]. Given their rarity within the bloodstream (a few CTCs among $10^9$ blood cells), the development of effective affinity-based isolation surfaces is imperative. These surfaces need to exhibit both high binding affinity towards CTCs and exceptional anti-fouling properties against white blood cells (WBCs)[27–29]. In this context, we have utilized a blend of experimental and simulation techniques to showcase the utilization of adaptable flexible M13 phage in constructing magnetic beads, forming a deformable surface that can tightly capture CTCs. This design serves to energetically enhance the capture of CTCs by allowing phage to twist to maximize the binding with the receptor on CTCs and entropically discourage the fouling of WBCs on this surface (Fig. 1a, b). The distinctive mechanical and structural attributes of the M13 phage impeccably fulfill the requirements for efficient CTC isolation (Fig. 1c).

Briefly, the bearing of many copies of pVIII protein on the sidewall of phage allows the phage-modified beads to furnish ample reactive sites for presenting numerous affinity ligands. Furthermore, the nanofiber's inherent flexibility permits it to twist freely, thus adjusting the ligand configuration to optimally align with the distribution pattern of target receptors on the CTC surface (Fig. 1c). This adaptive accommodation significantly enhances the multivalent binding interaction between the M13 phage and CTCs, consequently enabling highly efficient CTC capture. Moreover, owing to its low stiffness and Young's modulus—a quantification of its flexibility—the M13 phage possesses deformability within flowing solutions[30]. This property results in a heightened binding free energy that counters the non-specific adsorption of WBCs. This entropy-driven effect curbs the background signal from WBCs during CTC capture. Noteworthy is the fact that this phage-based physical shielding strategy for CTC capture obviates the need for additional coatings of anti-fouling polymers. This advantageous feature permits the optimal functionalization of the solid surface by affinity ligands, thereby augmenting target capture capabilities. Through this physical shielding approach, the deformable surface rooted in the phage structure offers a promising avenue for constructing efficient and anti-fouling surface bioassays.

## Results

### Phage engineering and its anchoring on Ni-IDA-grafted solid surface

The M13 phage was initially subjected to genetic engineering to introduce a 6His tag at the N-terminus of the pIII minor capsid protein. This process, depicted in Fig. 1a, Step (i), resulted in the creation of a modified phage nanofiber known as 6His-M13. The strategic placement of the 6His tag at the phage's tip played a pivotal role in enabling precise immobilization of the M13 phage onto surfaces such as glass slides or magnetic microbeads (MBs). These surfaces had been modified with Ni-iminodiacetic acid (Ni-IDA) and allowed for end-on immobilization (Fig. 2a, b).

To confer M13 with the ability to target CTCs, a CTC-specific aptamer known as S2.2, designed against the variable number tandem repeat (VNTR) region of Mucin 1 (MUC1)[31], was covalently linked to the exterior of the 6His-M13 phage. This linkage was achieved by introducing an azide functional group onto the phage's sidewall through a reaction involving NHS-PEG-N$_3$ and the N-terminal amine (NH$_2$) of the pVIII protein. This process, outlined in Fig. 1a, Step (ii), resulted in the creation of an azide-modified 6His-M13 phage termed N$_3$-M13, with an impressive grafting ratio of approximately 85.91% (Fig. 2c, d and Supplementary Fig. 1–2). Subsequently, the N$_3$-M13 phage was affixed to Ni-IDA MBs (Fig. 1a, Step (iii)), and then the dibenzocyclooctyne (DBCO)-labeled aptamer (DBCO-Apt) was efficiently linked to the phage's sidewall using a click reaction (Fig. 1a, Step (iv), Fig. 2e). This process yielded an assembly designated as an aptamer-flexible-M13-MB (A-f-M13-MB), with each individual phage carrying the modified aptamer (Apt-M13). An average of approximately 622 aptamers was found on the sidewall of each Apt-M13 (Supplementary Fig. 3). This distribution represented roughly a quarter of the pVIII proteins functionalized with aptamers, and there was an average distance of approximately 10–12 nm between neighboring aptamers[32]. Notably, considering the aptamer's size of about 2–3 nm, the density of aptamers on the M13 phage surface was well-arranged to prevent tangling or overlap between individual aptamers. We also evaluated the stability of A-f-M13-MB and found it could be preserved and stable in its lyophilized form for over ~7 weeks at −20 °C (Supplementary Fig. 4), underscoring its potential as a regular diagnostic tool.

### Mechanical properties and dynamic movement of the engineered M13 phage

Atomic force microscopy (AFM) analysis confirmed that the Apt-M13 phage displayed a notably low Young's modulus ($E$) and exhibited soft characteristics, consistent with the proteinaceous composition inherent in M13 phages, which are recognized as pliant materials (Fig. 3a, b). The tailored design of these phages, boasting high aspect ratios combined with their soft nature, facilitated pronounced twisting behavior within a solution (Fig. 3c), underscoring their inherent flexibility.

To comprehensively investigate the implications of M13 phage flexibility and dynamic transformation on ensuing CTC capture efficiency, deliberate measures were taken to enhance the rigidity of the M13 phage, thereby restraining its mobility within a solution. This involved treating the phage with 4% paraformaldehyde (PFA) or 100% ethanol (EtOH). As a result, the treated phages transformed into rigid M13 nanofibers, as evidenced by a substantial increase in their Young's modulus (2–3 times) and stiffness (~1.5–2 times) (Fig. 3a, b). Notably, these rigid M13 nanofibers exhibited a less convoluted morphology than their untreated, flexible counterparts (Fig. 3c–e). For better visualization of the flexibility difference between these phages at the moving state, a numerical simulation analysis was conducted to simulate the strain-induced deformation of M13 nanofibers with varying degrees of stiffness. The outcomes demonstrated that, under the same fluid shear stress, the deformation experienced by the

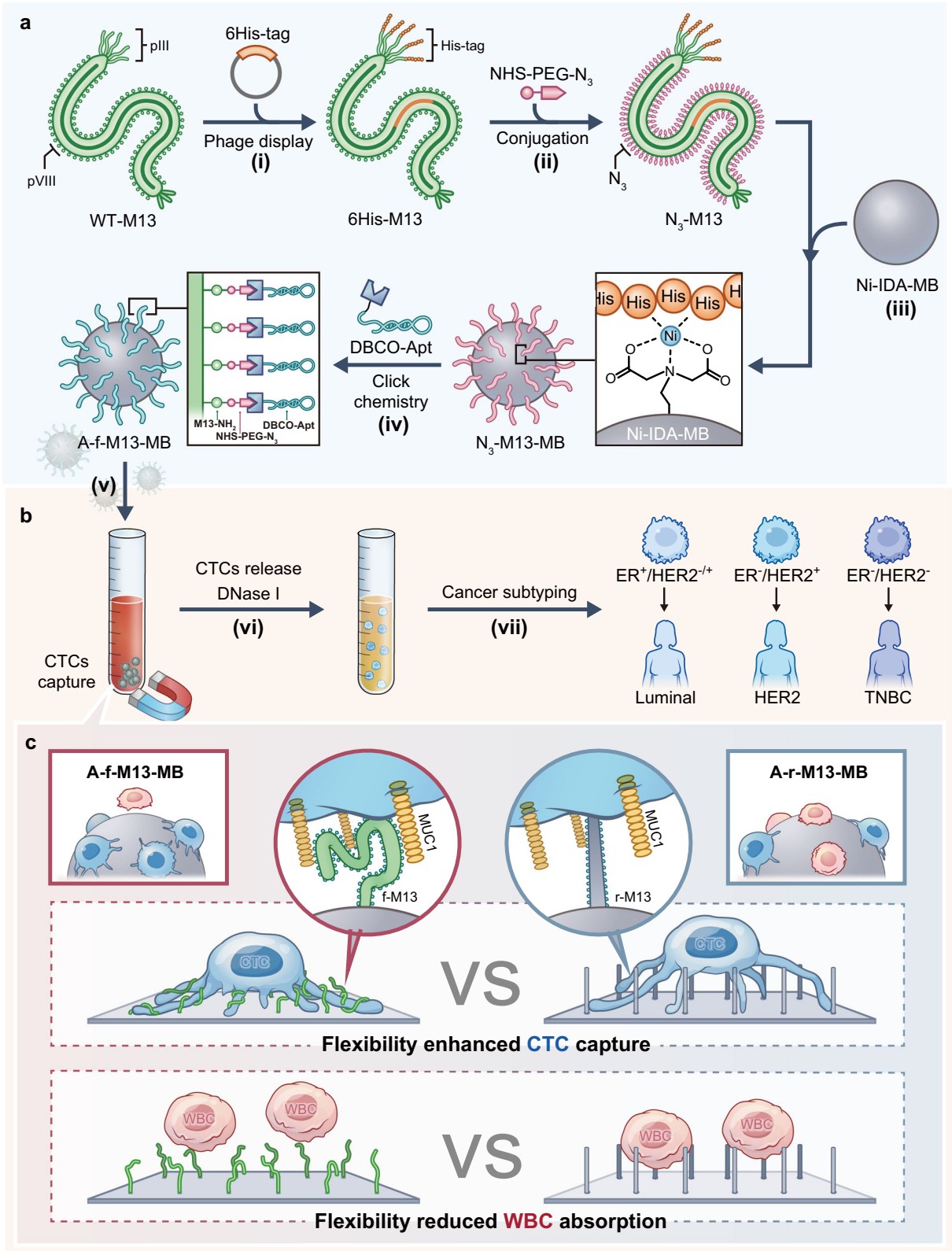

untreated, flexible M13 nanofibers was 1.65-1.74 times greater than that exhibited by the treated, rigid counterparts (Fig. 3f–h). This disparity signified a greater spatial configurational freedom within the flexible M13 nanofibers relative to their rigid counterparts.

Given that the PFA treatment rendered the M13 phages more rigid and did not impede subsequent aptamer decoration (i.e., sufficient $NH_2$- remained for aptamer decoration after PFA treatment, as indicated in Supplementary Table 2), we opted to refer to the PFA-treated M13 phages as "rigid M13" and maintained the nomenclature "flexible M13" for the untreated counterparts, for the sake of simplicity.

## Flexible M13 facilitates better CTC capture

The augmented affinity observed in multivalent receptor-ligand interactions is primarily attributed to the chelating and statistical

**Fig. 1 | Schematic illustration of the fabrication of the virus nanofiber-based deformable surface for highly efficient circulating tumor cell (CTC) isolation from whole blood. a** Engineering of the M13 virus nanofibers. Wild type M13 phage (WT-M13) was first genetically engineered to introduce a 6His tag at the N-terminus of the pIII minor capsid protein, generating 6His-M13 (Step (i)). The azide moiety was then introduced to the sidewall of the nanofibers by reacting NHS-PEG-$N_3$ with the N-terminal $NH_2$ of pVIII protein, producing $N_3$-M13 (Step (ii)). Thereafter, the $N_3$-M13 nanofiber was immobilized on the surface of Ni-IDA -grafted magnetic beads (Ni-IDA MB) in an end- on manner through the affinitive interaction between Ni and 6His, producing $N_3$-M13-MB (Step (iii)). Finally, the DBCO-labeled aptamer (DBCO-Apt) was conjugated with $N_3$-M13 via click reaction on $N_3$-M13-MB, forming Apt-M13-MB (Step (iv)), which was termed aptamer-flexible-M13-MB (A-f-M13-MB) to emphasize the flexibility of M13 phage. **b** The whole process of breast cancer CTC capture and profiling. Breast cancer CTCs in patients' whole blood were selectively captured by A-f-M13-MB and isolated via magnetic separation (Step (v)). The captured CTCs were released by DNase I and immuno-stained for the profiling of surface proteins including estrogen receptor protein (ER) and human epidermal growth factor receptor 2 (HER2) (Step) (vi)-(vii). CTCs stained as ER+/HER2+or-, ER-/HER2+, and ER-/HER2- were recognized as luminal subtype cell, HER2 positive sub-type, and basal-like subtype, respectively. **c** Flexibility enhanced CTC capture and reduced white blood cell (WBC) absorption. With low stiffness and Young's modulus, M13 phage exhibits a deformable feature with much configurational freedom in flowing solutions. This feature facilitated M13 on the A-f-M13-MB to adapt its configuration to promote the multivalent CTC-binding interactions to energetically enhance the CTC capture ability, whereas the rigid M13, formed by treatment of flexible M13 with paraformaldehyde (PFA), on the aptamer-rigid-M13-MB (A-r-M13-MB) could only offer limited sites for binding CTCs. Meanwhile, the flexibility feature of A-f-M13-MB led to the increase of the binding free energy for the non-specific adsorption of WBCs due to the entropy loss, thus guaranteeing a low WBC background.

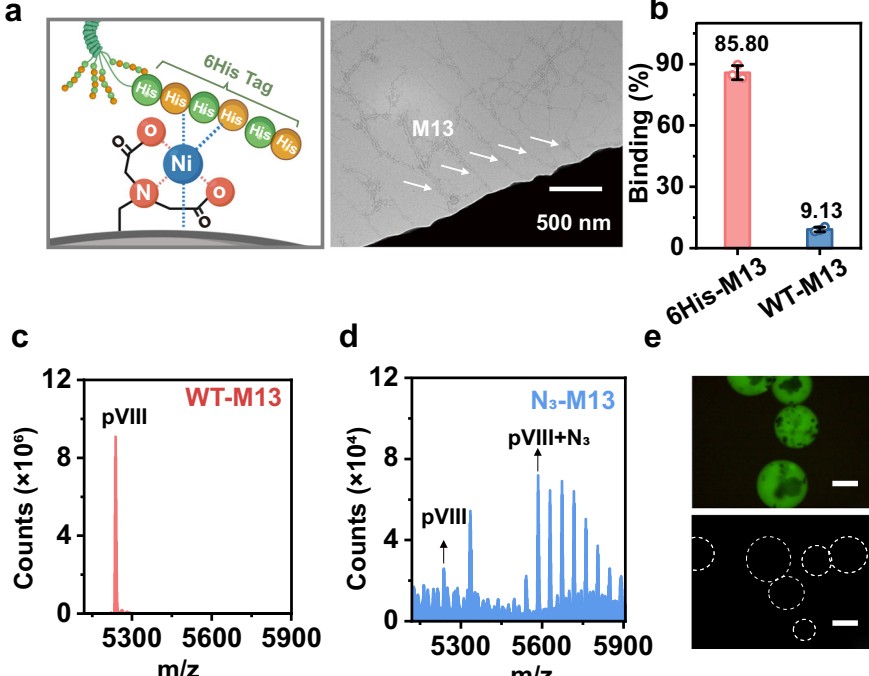

**Fig. 2 | Characterization of aptamer-displaying, flexible M13 nanofibers on magnetic beads (A-f-M13-MB). a** The cartoon illustration indicated the specific interaction between Ni-IDA on MBs and 6His tag on pIII of phage. Cartoon shown in Fig. 2a created with BioRender.com released under a Creative Commons Attribution-NonCommercial-NoDerivs 4.0 International license (https://creativecommons.org/licenses/by-nc-nd/4.0/deed.en). TEM images revealed that M13 nanofibers were anchored on MBs in an end-on manner. M13 nanofibers are indicated by arrows. **b** The much higher binding efficiency of 6His-M13 on Ni-IDA MBs compared to wild type (WT) M13 indicated the successful construction of 6His-M13 ($n = 3$ samples, mean ± s.d.). The Q-TOF LC/MS spectra of WT-M13 phage (**c**) and $N_3$-M13 phage (**d**). The peak at 5238 m/z confirmed the presence of pVIII in WT-M13, while the peak difference between 5238 m/z and 5585 m/z fitted the molecular weight of $N_3$-PEG, indicating presence of $N_3$-PEG. **e** Fluorescence microscopic images of FAM-A-f-M13-MB (upper) and control (lower). Aptamer was labeled with FAM, thus those MBs anchored with A-f-M13 emitted green fluorescence whereas the control MBs that anchored with WT-M13 didn't. Scale bar: 20 μm. Source data are provided as a Source Data file. WT-M13: wild type M13 phage; 6His-M13: M13 with 6His tag displayed on pIII; $N_3$-M13: 6His-M13 with $N_3$-PEG decorated on pVIII; Apt-M13: $N_3$-M13 with aptamer "clicked" onto pVIII.

rebinding effects. The potency of these effects largely hinges on the density and geometric arrangement of both receptors and ligands. This phenomenon has been extensively documented in previous studies[33-35]. In this particular investigation, MUC1 was chosen as the representative target receptor. MUC1 is a transmembrane glycoprotein that is often overexpressed in various epithelial cancers. Its extracellular domain extends around 200-500 nm beyond the cell surface, characterized by 20-120 VNTR repeats located in the N-terminal domain[36]. The elongated M13 nanofibers, spanning 880 nm and adorned with repeated VNTR-targeting aptamers, exhibit a favorable geometric fit with MUC1 (Fig. 4b). Although the distribution pattern might not achieve absolute alignment, the adaptable configuration of flexible M13 nanofibers effectively promotes the multivalent effect with the utmost efficiency.

By securing the flexible Apt-M13 nanofibers onto Ni-IDA MBs to forge A-f-M13-MBs, the binding affinity to CTCs was assessed using MCF-7 as a model MUC1+ CTC. Capitalizing on the molecular recognition between VNTR repeats and their corresponding aptamers, the sub-micron-scale fit between MUC1 and M13 nanofibers, and the micron-scale match between CTCs and M13-anchored topological interfaces (Fig. 4b and Supplementary Fig. 5a–c), our adaptable CTC-capturing A-f-M13-MB surface exhibited an extraordinary 238,000-fold increase in CTC binding affinity compared to that of the unattached aptamer (Fig. 4a). Furthermore, the CTC-binding affinity of the A-f-

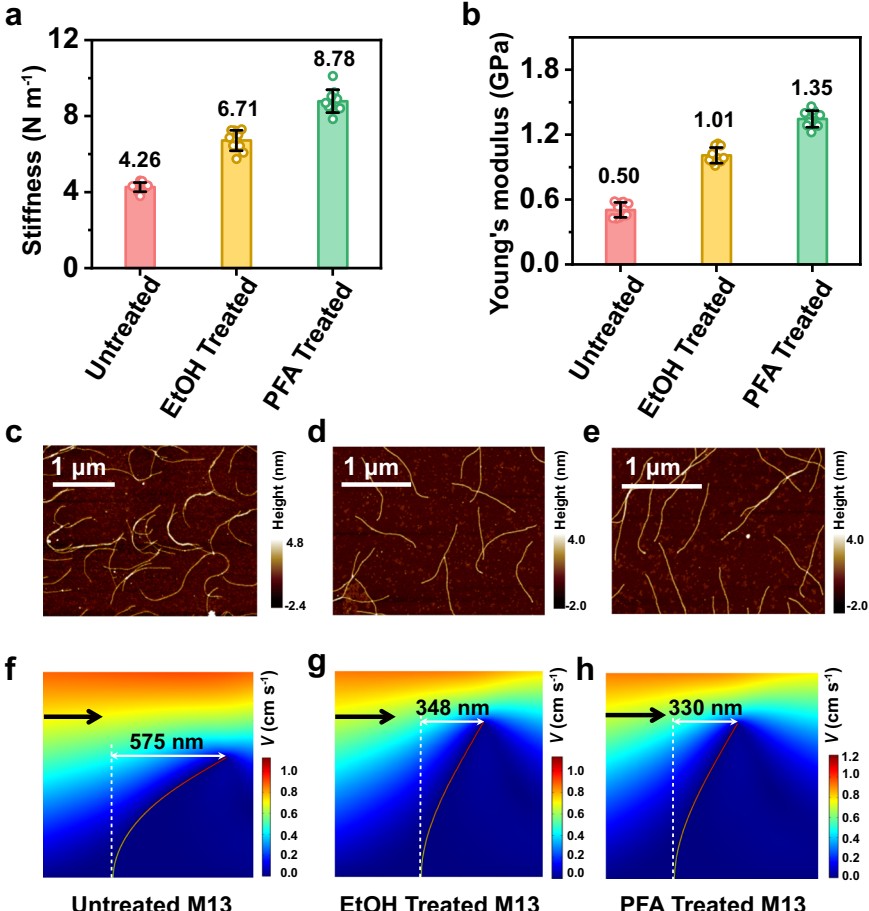

**Fig. 3 | Mechanical properties of M13 phage.** Stiffness (**a**) and Young's modulus (**b**) of untreated M13, EtOH-treated M13 and PFA-treated M13 measured by AFM under Hertzian mode ($n = 10$ samples, mean ± s.d.). M13 nanofibers were hardened to restrict their motion in solution by being treated with 100% ethanol (EtOH) or 4% paraformaldehyde (PFA) to produce EtOH-treated M13 and PFA-treated M13 for the comparison of mechanical properties. The stiffness of EtOH-treated M13 and PFA-treated M13 was 1.58-fold and 2.06-fold higher than that of untreated M13, whereas the Young's modulus of EtOH-treated M13 and PFA-treated M13 was 2.02-fold and 2.70-fold higher than that of untreated M13, respectively. AFM images of untreated M13 (**c**), EtOH-treated M13 (**d**) and PFA-treated M13 (**e**). AFM images revealed that these treated rigid M13 presented a less twisty morphology than untreated M13. The loading amount of three types of M13 phages was identical ($10^9$ pfu). Numerical simulations results for untreated M13 (**f**), EtOH-treated M13 (**g**) and PFA-treated M13 (**h**) under a flow field, wherein the arrow indicates the flow direction. Subjected flow speed: 2 cm/s. When M13 was subjected to the same force from fluid shear stress, the deformation of untreated M13 was 1.74 and 1.65 folds larger than that of EtOH-treated M13 and PFA-treated M13, respectively. Source data are provided as a Source Data file. EtOH-treated M13: M13 phage treated with 100% ethanol (EtOH); PFA-treated M13: M13 phage treated with 4% paraformaldehyde (PFA).

M13-MB surface exceeded that of the surface formed by directly immobilizing the aptamer onto MBs (i.e., A-MB) by approximately 19,200-fold. This underscores the pivotal role of the flexible Apt-M13 phage. Interestingly, substituting the flexible Apt-M13 phage with its rigid counterpart resulted in A-r-M13-MBs (A for "aptamer," r for "rigid") that displayed a substantial 22.8-fold decrease in CTC-binding affinity. Given that the aptamer loading level remained consistent (as shown in Supplementary Table 2) and a comparable topological enhancement effect was observed for both A-f-M13-MBs and A-r-M13-MBs (Supplementary Fig. 5a−c), it becomes evident that the flexibility of Apt-M13 is the primary driver behind the heightened CTC-binding affinity.

To delve deeper into the role of M13's flexibility in influencing the adhesive force between the CTC population and the affinity solid surface, we conducted a cell adhesion assay based on centrifugation[37,38]. In this setup, either the flexible or rigid $N_3$-M13 was affixed to Ni-IDA glass slides and equipped with aptamers (Supplementary Fig. 6, 7). After a 30-minute cell capture period, a reverse centrifugal force was applied to the M13 slides with attached cells. The results depicted in Fig. 4c, d unveiled that the flexible M13 exhibited a minimum of 18% higher adhesion force than its rigid counterpart. This outcome underscores once again that the inherent flexibility of M13 is instrumental in achieving robust binding to receptors on CTCs.

To gain deeper physical insights into the augmented multivalent interaction owing to flexibility, we employed dissipative particle dynamics (DPD) simulations[39] to explore the intricate molecular-level interactions between CTCs and MBs (see the simulation methods and modeling details in the Supplementary Information). In the simulations, three distinct types of MBs, namely A-MB, A-f-M13-MB, and A-r-M13-MB, were constructed (Supplementary Fig. 8). As illustrated in Fig. 4e, CTCs could be captured onto all three MB surfaces; however, the underlying microscopic interaction modes differed significantly. For instance, in the case of A-MB, an individual CTC receptor could interact with only one or two aptamers located near the binding site. Conversely, in the case of A-r-M13-MB, due to the presence of multiple aptamers on each M13 phage, a single CTC receptor could simultaneously engage with numerous aptamers. The most intricate scenario unfolded with A-f-M13-MB, where the flexibility of the M13 phage allowed it to conform or twist to accommodate the receptor distribution on CTCs. Consequently, a CTC receptor could concurrently

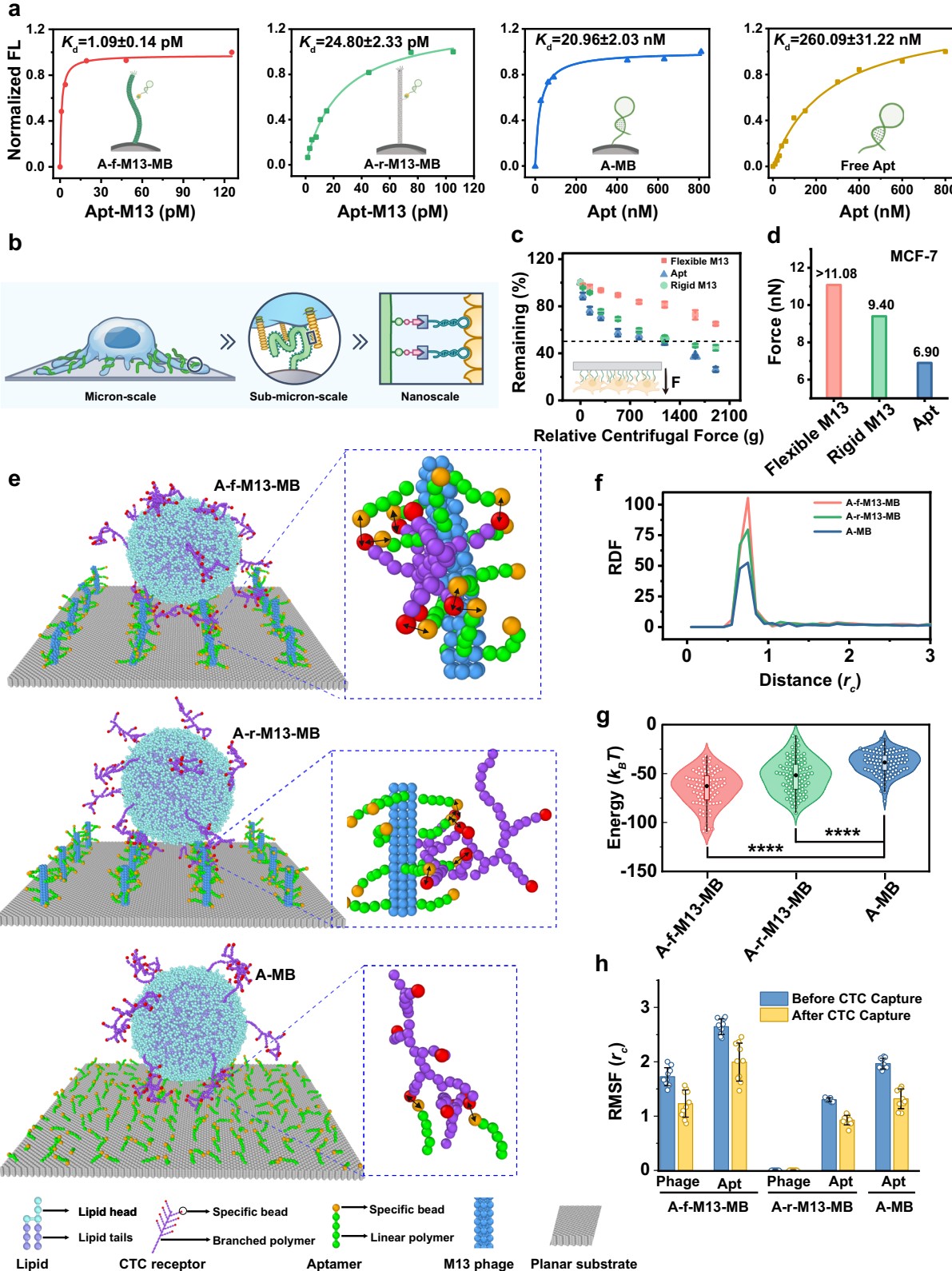

interact with a significantly larger number of aptamers. The micro-level distinction in interactions was further manifested in the radial distribution function (RDF) profiles. Specifically, the RDF peak values were approximately 105, 79, and 52 for A-f-M13-MB, A-r-M13-MB, and A-MB, respectively (Fig. 4f). The total energy of receptor-aptamer interaction exhibited an ascending order from A-f-M13-MB (-65 $k_BT$), to A-r-M13-MB (-52 $k_BT$), and then to A-MB (-−39 $k_BT$) (Fig. 4g). This sequence

indicates that the interaction between CTCs and A-f-M13-MB was the most energetically favorable, attributable to the increased contacts between the M13-anchored aptamers and CTC receptors compared to the other two MB types.

It's worth noting that the freedom of M13 phage and/or the aptamers was restricted upon CTC adsorption on the MB surface (Fig. 4h). This reduction increased the entropy term (-$T\Delta S$) for the

**Fig. 4 | Flexible M13 enhanced CTCs capture performance. a** The dissociation constants ($K_d$) of A-f-M13-MB, A-r-M13-MB, A-MB and free aptamer. The $K_d$ value reflects the affinity between CTCs and affinity ligands, with a lower $K_d$ corresponding to a higher CTC binding affinity. **b** Illustration of the molecular matching between VNTR repeats in MUC1 and aptamers on Apt-M13, sub-micron-scale matching between MUC1 (200–500 nm in size) and M13 nanofibers (880 nm long), and micron-scale matching between CTCs and M13 nanofiber-anchored topological interfaces. **c** Percentages of cells remaining on the slide under different relative centrifugal forces ($n = 3$ samples, mean ± s.d.). **d** The detachment forces required for separating MCF-7 cells from different slides' surface calculated from centrifugation-based cell adhesion assay indicated that the CTC binding force (MCF-7 as CTC model) was decreasing in the order of flexible M13 > rigid M13 > aptamer. **e** Representative final snapshots of different MB surfaces interacting with CTCs in the simulations. The arrows indicate the aptamer-receptor binding sites. **f** Radial distribution function (RDF) of aptamer-specific-bead/receptor-specific-bead pairs in the three cases. RDF indicates the relative density of aptamers around receptors, which was decreased in the order of A-f-M13-MB > A-r-M13-MB > A-MB.

**g** Total energy of receptor-aptamer interaction was increased in the order of A-f-M13-MB < A-r-M13-MB < A-MB, as more contacts between the specific beads constituting the aptamers and the receptors resulted in lower total energy of the receptor-aptamer interaction ($n = 101$ tests). Unpaired two-sided Student's $t$ test. The central dot is the median; box bounds are 25th and 75th percentiles, upper and lower limits of whiskers are 1.5 × interquartile ranges. Values outside of the upper and lower limits are defined as outliers. $*p < 0.05$, $**p < 0.01$, $***p < 0.001$, $****p < 0.0001$. **h** The root mean square fluctuation (RMSF) of the aptamer and/or the M13 of the three MBs before and after the CTC adsorption ($n = 10$ tests, mean ±s .d.). Source data are provided as a Source Data file. Cartoons shown in (**a**) and (**c**) were created with BioRender.com released under a Creative Commons Attribution-NonCommercial-NoDerivs 4.0 International license (https://creativecommons.org/licenses/by-nc-nd/4.0/deed.en). A-MB: aptamer-modified-magnetic beads; A-f-M13-MB: Ni-IDA-MBs anchored with aptamer-modified-flexible M13; A-r-M13-MB: Ni-IDA-MBs anchored with aptamer-modified-rigid M13 (PFA-treated M13 as a rigid M13 model); Apt aptamer.

interaction, creating an entropic deterrent to adsorption. However, given the pronounced strength of the specific receptor-aptamer interaction, the interaction energy prevailed over these entropic factors during the adsorption process. This dynamic favored the interaction by causing a decline in the energy term that outweighed the concurrent rise in the entropy term.

### Twisty M13 facilitates less WBC adsorption

The exceptional scarcity of CTCs demands surfaces with both high CTC binding affinity and robust anti-fouling capabilities. To assess the anti-fouling performance of our engineered surfaces, we employed human Burkitt's lymphoma Ramos cells as a model for white blood cells (WBCs). Both flexible and rigid M13 nanofibers, each bearing the same level of aptamers, were anchored to Ni-IDA MBs. These MBs were subsequently exposed to $10^6$ Ramos cells. Notably, the outcomes revealed that only about ~1354 WBCs adhered to the A-f-M13-MBs following a 30-minute incubation. In stark contrast, the number of WBCs binding to A-r-M13-MBs and A-MBs was 7.5-fold and 8.9-fold higher, respectively (Fig. 5a). Correspondingly, the centrifugation-based cell adhesion assay disclosed a substantially reduced WBC binding force of 8.64 pN for A-f-M13-MBs. Remarkably, this force accounted for merely one-third and one-sixth of the binding forces observed for A-r-M13-MBs and A-MBs, respectively (Fig. 5b, c).

For deeper insight into the distinct anti-fouling characteristics of the three surfaces at the molecular level, we employed DPD simulations. The modeled WBC, as shown in Fig. 5d, demonstrated a propensity for non-specific adsorption onto all three MB surfaces. Given the absence of specific receptor-ligand interactions, the contact between WBCs and MBs was significantly weaker, with the RDF peak registering at approximately 2–3 (Fig. 5e) compared to the more pronounced CTC-MB contacts (peaking within the 50-105 range, as seen in Fig. 4f). Notably, the RDF peak for A-MB was the highest, while the interaction energy was the lowest (Fig. 5f), thus rendering WBC adsorption energetically most favorable for A-MB. On the other hand, the RDF peak and interaction energy were nearly identical for A-f-M13-MB and A-r-M13-MB, implying similar energetic probabilities for WBC adsorption. However, the freedom of both M13 phage and ligands was restricted upon WBC adsorption to the MB surface (Fig. 5g). Importantly, for A-f-M13-MB, both M13 phage and aptamer freedoms were diminished, while for A-r-M13-MB, only aptamer freedom was curtailed. Consequently, the entropy term ($-T\Delta S$) was higher for A-f-M13-MB than for A-r-M13-MB. Of greater significance, given the relatively weak non-specific interaction (manifesting as a total interaction energy of several $k_BT$), the entropy term carried more weight than the energy term in dictating the interaction free energy. This dynamic resulted in less entropically favorable WBC adsorption onto the A-f-M13-MB surface than A-r-M13-MB.

### Efficient isolation of CTCs by M13-anchored MBs (M13-MBs)

Having elucidated the underlying mechanism behind the flexibility-driven high-affinity binding and effective prevention of bio-fouling, we proceeded to evaluate the performance of aptamer-anchored flexible M13-MBs (A-f-M13-MB) for CTC isolation under physiological conditions (Fig. 1a, Step (v)). Following the confirmation of the aptamer's targeting capability towards MUC1-positive cells (Supplementary Fig. 9), we gauged the selectivity of A-f-M13-MB for CTC capture by incubating it with various cancer cells for 30 minutes (Fig. 6a, Supplementary Fig. 10, 11). Notably, as illustrated in Fig. 6a, A-f-M13-MB successfully captured over 92.21% of MUC1 positive cancer cells (MCF-7, A549), whereas the capture efficiencies for negative cells (HepG2, SK-Hep-1) were markedly lower, falling below 13.01%. This discrepancy underscores the impressive CTC-capture selectivity achieved by A-f-M13-MB. Furthermore, the efficiency of CTC capture by A-f-M13-MB remained scarcely affected in whole blood conditions, maintaining a capture rate exceeding 90.25% (Fig. 6a). Typically, aptamers are susceptible to degradation by nucleases present in whole blood, which significantly hampers their isolation performance in clinical settings. In our case, it is possible that the numerous aptamers on M13 nanofibers have the capacity to replenish degraded aptamers nearby, and they might dynamically adjust their positions by freely twisting the M13 scaffold. Consequently, this mechanism upholds high capture efficiency through multivalent binding, facilitated by a statistically rebinding mechanism.

We further compared the CTC capture and release efficiency of A-f-M13-MB, A-r-M13-MB, and A-MB using MCF-7 as the model cancer cell. Initially, MCF-7 cells were incubated with A-f-M13-MB for 30 min, magnetically isolated, rinsed, and subsequently released through DNase I digestion for microscopic enumeration (Fig. 1a, Step (vi)). As depicted in Fig. 6b, A-f-M13-MB achieved a capture efficiency of 91.46% for MCF-7 cells, which was approximately 12.08% higher than A-r-M13-MB and 24.67% higher than A-MB. Simultaneously, the flexible twisting of M13 nanofibers afforded sufficient interspace for DNase I access, resulting in a 23.74% improvement in cell release efficiency for A-f-M13-MB compared to A-r-M13-MB. This discrepancy in CTC release efficiency was even more pronounced when comparing A-f-M13-MB to A-MB (83.85% for A-f-M13-MB versus 49.73% for A-MB). In the case of A-MB, where aptamers were directly immobilized on the surface, DNase I struggled to reach the aptamer binding site once it was blocked post-CTC binding. Furthermore, CTC release performance remained nearly identical between whole blood samples and buffer solutions (Supplementary Figs. 12, 13), ensuring practical applicability in clinical scenarios.

Maintaining the purity and viability of released CTCs is paramount for downstream analysis. The flexible M13 nanofibers exhibit

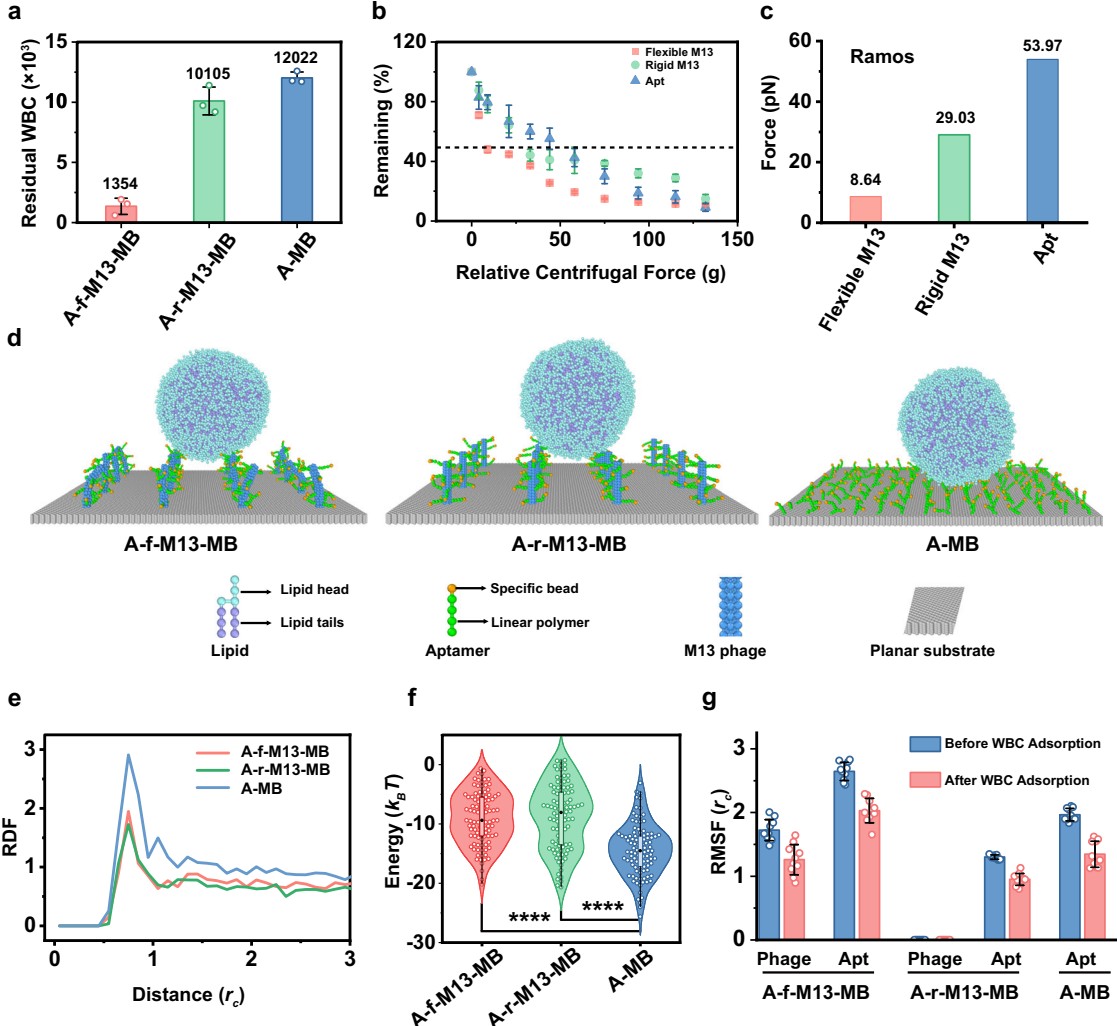

**Fig. 5 | Flexible M13 reduced non-specific absorption of WBCs. a** Number of WBCs adsorbed on the A-f-M13-MB, A-r-M13-MB and A-MB after 30-min incubation ($n = 3$ samples, mean ± s.d.). **b** Percentages of residual WBCs on the slide under different relative centrifugal forces ($n = 3$ samples, mean ± s.d.). Either flexible or rigid $N_3$-M13 were anchored on Ni-IDA glass slides and loaded with aptamers. WBCs were first incubated with M13 slides for 30 min, and a reversed centrifugation force was then applied on the WBC-attached slide for 5 min. **c** The detachment forces required for separating WBCs from the surface of different slides calculated from the centrifugation-based cell adhesion assay indicated that the WBC binding force (Ramos cell as a model) was increased in the order of flexible M13<rigid M13 <aptamer. **d** Representative final snapshots of different MB surfaces interacting with WBCs in the simulations. The results indicated that WBCs could be adsorbed onto all three MB surfaces. **e** Radial distribution function (RDF) of aptamer-specific-bead/receptor-specific-bead pairs in the three cases. RDF indicates the relative density of aptamers around receptors. Higher RDF corresponds to a lower total resistance to WBC attachment, leading to a relatively higher log10-

energy of the receptor-aptamer interaction. **f** Total energy of receptor-aptamer interaction in three cases ($n = 101$ tests). Unpaired two-sided Student's $t$ test. The central dot is the median; box bounds are 25th and 75th percentiles, upper and lower limits of whiskers are 1.5 × interquartile ranges. Values outside of the upper and lower limits are defined as outliers. $*p < 0.05$, $**p < 0.01$, $***p < 0.001$, $****p < 0.0001$. Results from (**e**) and (**f**) revealed the adsorption of WBCs on A-MB was the most energy-favorable, compared to the reduced adsorption of WBCs on A-f-M13-MB and A-r-M13-MB. ($n = 101$ tests). **g** The root mean square fluctuation (RMSF) of the aptamer and/or the M13 of the three MBs before and after the WBC adsorption, indicating that the adsorption of WBCs onto the A-f-M13-MB surface is the least entropy-favorable ($n = 10$ tests, mean ± s.d.). Source data are provided as a Source Data file. A-MB: aptamer-modified-magnetic beads; A-f-M13-MB: Ni-IDA-MBs anchored with aptamer-modified-flexible M13; A-r-M13-MB: Ni-IDA-MBs anchored with aptamer-modified-rigid M13 (PFA-treated M13 as a rigid M13 model); Apt aptamer.

depletion index of WBCs for A-f-M13-MB than for A-r-M13-MB and A-MB (Fig. 6c). Since most WBCs are adsorbed non-specifically onto MBs without selective capture by aptamers, the DNase I-based strategy exclusively releases CTCs. This contributes to an increased WBC depletion index through the selective release of CTCs. Additionally, DNase I hydrolyzes the aptamers rather than degrading cells, ensuring high viability of the released CTCs, as confirmed by PI/AO staining (Fig. 6d, e, viability >96.29%). The isolated CTCs were successfully cultivated in 96-well plates and demonstrated steady adherence to the well surface over a 36-h period (Fig. 6f). We also evaluated the migration ability and the phenotype drifting profile after the CTCs

were released and re-cultured. As shown in Supplementary Fig. 14, the expression level of estrogen receptor protein (ER, feature biomarker of MCF-7 cells) showed no difference after CTC isolation and re-culture, indicating no phenotype drifting occurred after treatment with A-f-M13-MB. Namely, the isolated CTCs can faithfully represent the phenotypic feature of the original ones. From the analysis results indicated in Supplementary Fig. 14, it is clear that after isolation and re-culture, the isolated CTCs have comparable or even better migration ability compared to the original CTCs. The high viability of the isolated CTCs can well meet the requirement for downstream applications including in vitro CTCs culturing or mouse transplantation xenograft construction.

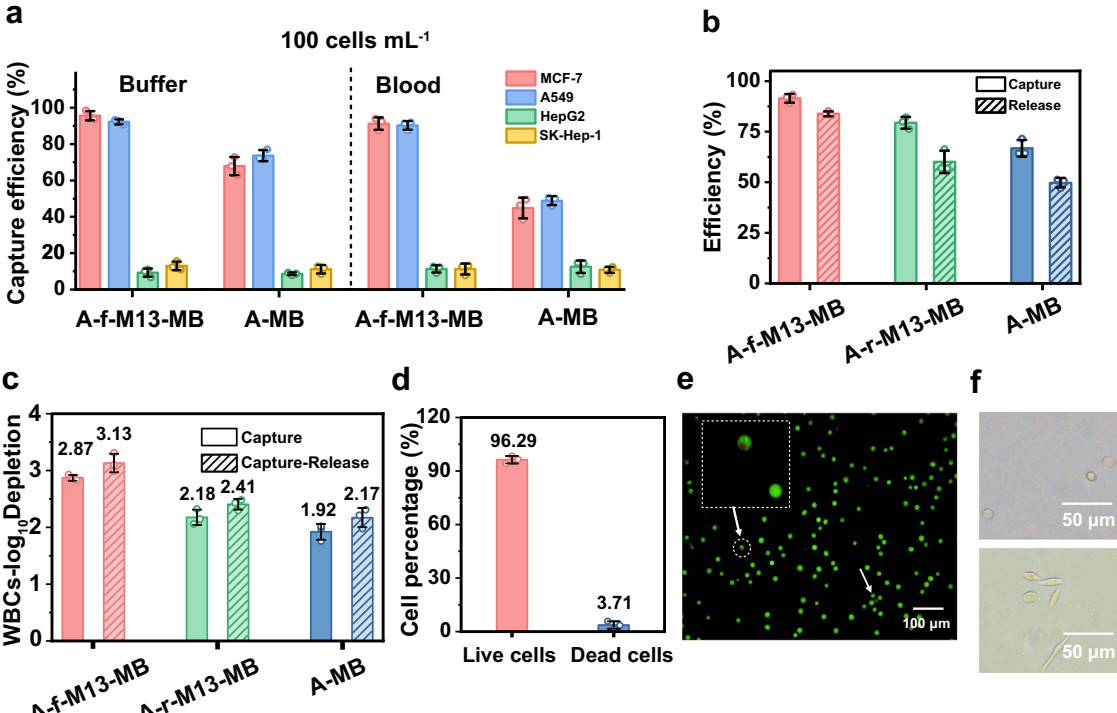

**Fig. 6 | Capture and release performance of A-f-M13-MB. a** Capture efficiency of MUC1-positive cancer cells (MCF-7 and A549) and MUC1-negative cancer cells (HepG2 and SK-Hep-1) using A-f-M13-MB and A-MB in buffer and blood, respectively ($n = 3$ samples, mean ± s.d.). The results indicated that A-f-M13-MB can effectively capture the MUC1 positive cancer cells (MCF-7, A549), but barely adsorbed negative cells (HepG2, SK-Hep-1) both in buffer and whole blood. **b** Capture and release performance of A-f-M13-MB, A-r-M13-MB and A-MB towards MCF-7 cells ($n = 3$ samples, mean ± s.d.). **c** Depletion index value of WBCs with A-f-M13-MB, A-r-M13-MB and A-MB ($n = 3$ samples, mean ± s.d.). The depletion index value reflects the purity of CTCs after capture and release, with a higher $\log_{10}$-depletion index value indicating a higher CTC purity. The results revealed a higher CTC purity after A-f-M13-MB capture compared to other two MBs, as well as an improved CTC purity

after CTC release. **d** Viability of the isolated MCF-7 cells calculated from PI/AO staining results ($n = 3$ samples, mean ± s.d.). **e** Representative fluorescence microscope images of the isolated live cells and dead cells. The released CTCs were stained by PI/AO, and the live cells emitted green fluorescence and dead cells (indicated by arrows) emitted red fluorescence. Scale bar: 100 μm. Insert is the enlarged dead cells. **f** Bright-filed images of the released cells (upper) and the cells after being further cultured for 36 h (lower). The images revealed the released cells were well alive and could be adhered on the well surface 36 h after in vitro cultivation. Source data are provided as a Source Data file. A-MB: aptamer-modified-magnetic beads; A-f-M13-MB: Ni-IDA-MBs anchored with aptamer-modified-flexible M13; A-r-M13-MB: Ni-IDA-MBs anchored with aptamer-modified-rigid M13 (PFA-treated M13 as a rigid M13 model).

## Clinical specimen analysis

Breast cancer (BC) stands as the most prevalent malignant tumor and ranks second in terms of cancer-related mortality among women globally[40]. This disease exhibits significant heterogeneity, categorized into distinct subtypes including luminal, human epidermal growth factor receptor 2 (HER2)-positive, and basal-like subtypes. Notably, patients with the luminal subtype generally experience a more favorable prognosis than the other subtypes[41-43]. Differential BC subtypes necessitate tailored treatments, suggesting the urgency for the development of a rapid yet precise early-stage subtyping diagnostic approach.

MUC1, excessively overexpressed in over 90% of breast tumors and 60% of captured circulating tumor cells (CTCs) from metastatic breast, lung, pancreatic, and colon cancer patients, holds promise as a biomarker for BC CTCs[44-46]. We embarked on assessing the feasibility of isolating BC CTCs using A-f-M13-MBs. Before translating to clinical samples, it is vital to evaluate the capturing efficacy of A-f-M13-MBs using model cells representing these BC subtypes. Immunofluorescence staining confirmed MUC1+/HER2-/estrogen receptor protein (ER)+ MCF-7 cells (Supplementary Fig. 15) as luminal subtypes, MUC1+/HER2+/ER- SK-BR-3 cells as HER2-positive subtypes (Supplementary Fig. 16), MUC1+/HER2-/ER- MDA-MB-231 cells as basal-like subtypes (Supplementary Fig. 17), and MUC1-/HER2-/ER- MCF-10A cells as normal breast mammary epithelial cells (Supplementary Fig. 18). Since these model BC cells were MUC1-positive, A-f-M13-MB

consistently captured them with efficiencies ranging from 83.34% to 93.51% (Fig. 7a). However, non-specific capture of the normal cell line MCF-10A was a mere 6.99%, indicating the effective CTC selectivity of A-f-M13-MB in distinguishing BC cells from normal breast epithelial cells (Fig. 7a). Given that ER/HER2 expression significantly varies across subtypes, subtyping could be logically confirmed through immunofluorescence staining.

Having established nearly identical capture efficiency across different BC subtype CTCs, we proceeded to assess the diagnostic efficacy of A-f-M13-MB by analyzing blood samples from 56 BC patients, 34 benign patients, and 10 healthy volunteers (Supplementary Data 1) without any prior treatment. Each assay involved evaluating 0.5 mL of whole blood, followed by CTC capture/release processes and a standard three-color immunocytochemistry staining protocol[47]. Cells exhibiting CD45 negativity, DAPI positivity, and CK positivity (DAPI+/CK+/CD45-) were recognized as CTCs, while those staining positive for CD45 and DAPI but negative for CK (DAPI+/CK-/CD45+) were identified as white blood cells (WBCs) (Supplementary Fig. 19). Worth mentioning is our EpCAM-independent isolation strategy, which involves recognizing MUC1, thus facilitating the capture of EpCAM-negative CTCs from metastatic BC patients (Patient No. 16, No. 72 and No. 88, Supplementary Fig. 19 and Supplementary Data 1). This approach enhances clinical utility, especially for EpCAM-negative CTCs undergoing epithelial-mesenchymal transition (EMT) that often exhibit heightened metastatic potential. Patient No. 16, 72 and 88's severe

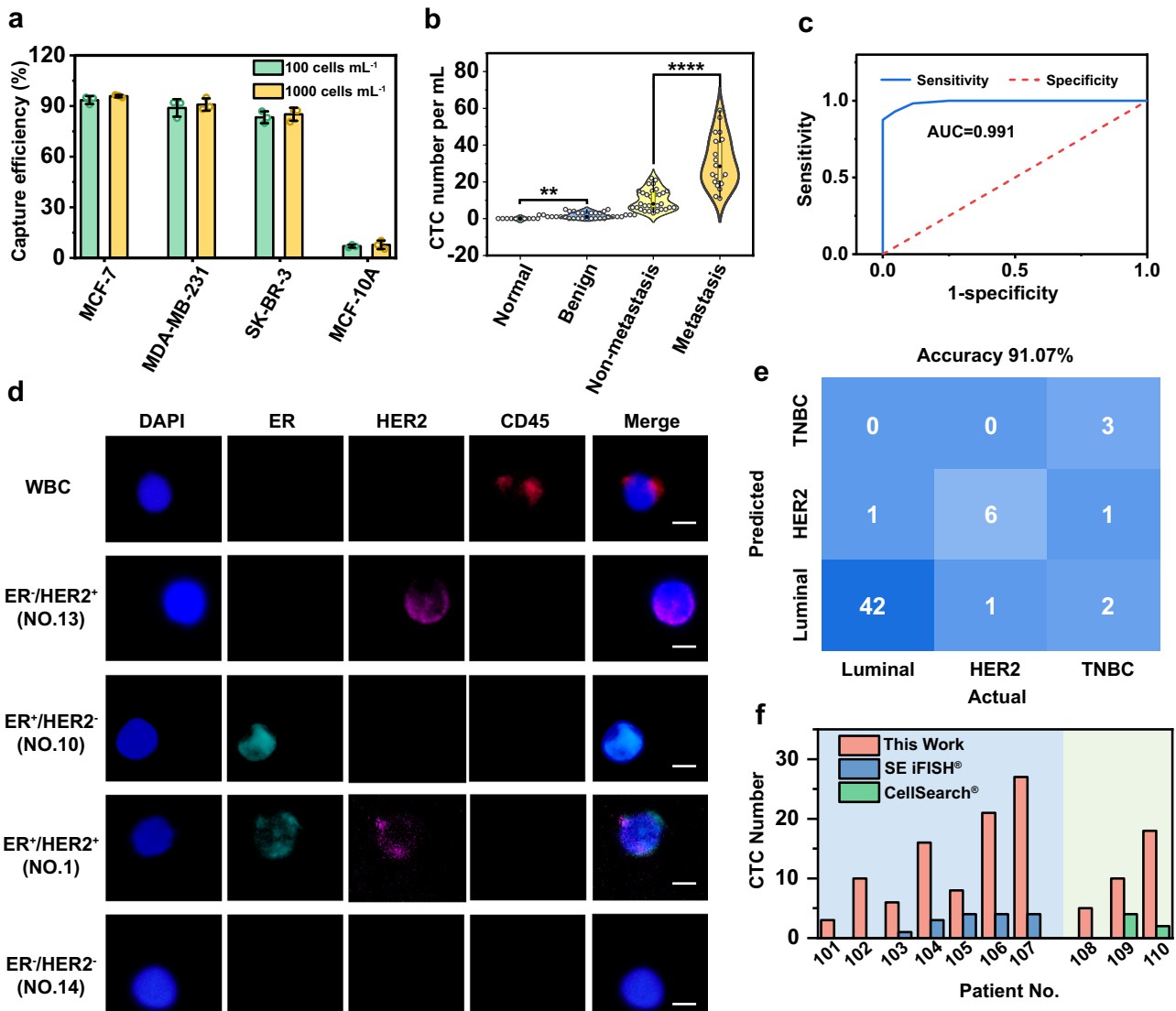

**Fig. 7 | CTC isolation and diagnosis performance in clinical applications. a** The capture efficiency of different breast cell lines indicated that the CTC affinity interfaces could selectively capture several subtypes of breast cancer cells but not normal breast mammary epithelial cells ($n = 3$ samples, mean ± s.d.). **b** Distinct difference in the enumeration of CTCs in 1 mL blood samples from healthy volunteers ($n = 10$), benign patients ($n = 34$), non-metastatic breast cancer (BC) patients ($n = 32$), and metastatic BC patients ($n = 24$). Unpaired two-sided Student's *t* test. The central dot is the median; box bounds are 25th and 75th percentiles, upper and lower limits of whiskers are 1.5 × interquartile ranges. Values outside of the upper and lower limits are defined as outliers. \**p* < 0.05, \*\**p* < 0.01, \*\*\**p* < 0.001, \*\*\*\**p* < 0.0001. **c** ROC analysis of CTC numbers between the cancer patient groups and healthy/benign groups. The receiver operating characteristic (ROC) curve showed good diagnostic performance (AUC = 0.991) in differentiating cancer patient groups and healthy/benign groups. **d** Immunofluorescence staining

(DAPI/ER/HER2/CD45) of CTCs isolated from patients. Scale bar: 10 μm. CD45 was the biomarker for WBC and CK was the biomarker for CTC, with DAPI$^+$/CK$^+$/CD45$^-$ cells recognized as CTCs and DAPI$^+$/CK$^-$/CD45$^+$ cells identified as WBCs. ER and HER2 were used for the profiling of three subtypes of BC CTCs, including luminal (HER2$^{+ or -}$/ER$^+$), HER2-positive (HER2$^+$/ER$^-$) and triple negative breast cancer (TNBC, belongs to basal-like subtype) (HER2$^-$/ER$^-$). 51 out of 56 molecular profiling results were in accordance with clinical diagnostic results. **e** Confusion matrix showing the subtyping accuracy of luminal, HER2, and TNBC by molecular profiling of CTCs. **f** Comparison of the CTC numbers isolated by A-f-M13-MB and CellSearch® or SE iFISH® in blood revealed a higher CTC capture ability and a better isolation performance of A-f-M13-MB in clinical applications. CellSearch® and SE iFISH® collected CTC from 7.5 mL and 6.0 mL blood, respectively, whereas our approach isolated CTC from 1.0 mL blood. A-f-M13-MB: Ni-IDA-MBs anchored with aptamer-modified-flexible M13; Source data are provided as a Source Data file.

lymphatic metastasis situation (Supplementary Data 1) particularly underscores this point. As illustrated in Fig. 7b and Supplementary Data 1, CTCs were successfully identified and isolated from all BC patients, ranging from 3 to 59 CTCs per 1 mL of blood. In contrast, no more than 5 CTCs were found in blood samples from healthy donors or benign patients. CTC enumeration also facilitated differentiation between metastatic and non-metastatic BC patients. The receiver operating characteristic (ROC) curve exhibited promising diagnostic performance, with a sensitivity of 92.86%, specificity of 95.45%, and an area under the curve (AUC) of 0.991, with an optimal cutoff value of >4

CTC mL$^{-1}$ (Fig. 7c, Supplementary Data 2). Among the 56 patients sharing the same clinical diagnosis, the molecular profiling of isolated CTCs was analyzed by assessing ER and HER2 expression levels (Fig. 7d). As presented in Fig. 7e, 51 out of 56 patients were accurately classified into the correct subtype, achieving a diagnostic accuracy of 91.07%. Two benchmark methods, the only FDA-approved CellSearch® method and another prevalent SE iFISH® technique, were also compared with this approach in terms of isolation capability (Fig. 7f and Supplementary Table 3). Thanks to the adaptive fitting multivalent binding of M13, our approach demonstrated a significant increase in

the number of isolated CTCs compared to CellSearch® and SE iFISH®. This enhancement significantly reduces the misdetection rate and false negatives associated with A-f-M13-MB.

## Isolation of CTCs with different EMT subphenotypes by M13-MBs

The existence of various CTCs with different EMT subphenotypes in cancer patient's blood, especially those suffering from metastasis, highlighted the importance to develop efficient CTC capture interfaces covering different EMT subphenotypes. We thereby constructed a Y-shaped DNA scaffold to replace the MUC1 aptamer, to better isolate CTCs with different EMT characteristics. The Y-shaped DNA scaffold was assembled by an anti-vimentine aptamer[48] (targeting the M-type CTC) and an anti-EpCAM aptamer[49] (targeting the E-type CTC) (see Supplementary Fig. 20a, b for assembling detail, and see Supplementary Table 1 for DNA sequences), and then attached to M13-MB via the same procedure described in Methods section "Aptamer-loading on N$_3$-M13 phage-immobilized magnetic microbeads (MBs)".

For evaluating the CTC isolation performance of the resultant A-f-M13-MB, the E-type CTC model cell (MCF-7 cell, EpCAM$^+$, N-cadherin$^-$) and M-type CTC model cell (MDA-MB-231 cell, EpCAM$^-$, N-cadherin$^+$) were mixed with different ratios and spiked in the CTC-free whole blood, followed by the isolation with A-f-M13-MB and immuno-fluorescence staining for phenotyping. As shown in Supplementary Fig. 20c, the CTC mixtures maintained a high capture efficiency regardless of the ratio of E-type/M-type, and the subphenotypes can be clearly identified by immunofluorescence images (Supplementary Fig. 20d). This underscored the ability of A-f-M13-MB to isolate and identify CTCs with different EMT characteristics.

## Discussion

This study reveals the significant role played by the mechanical attributes of virus-modified solid surfaces in the isolation of rare cells, such as CTCs, with respect to both target cell binding affinity and the mitigation of non-specific interactions with non-target cells like WBCs. The inherent flexibility and deformability of virus nanofibers confer a self-adjusting capability to the solid surface, thereby amplifying multivalent interactions between the aptamers residing on the nanofibers and the receptors on target cells through energy-driven processes. Additionally, these flexible virus nanofibers on the solid surface yield a marked reduction in the non-specific adsorption of non-target cells when compared to their rigid counterparts, attributed to an entropy-driven mechanism resulting from the greater loss of freedom within flexible nanofibers relative to rigid ones. Our approach, reliant on flexible virus nanofibers, outperforms conventional methods based on anti-fouling polymers, as it avoids the potential drawbacks linked to such polymers and simultaneously provides the necessary space on the solid surface for the specific ligand recognition of target cells. This phage-based strategy not only introduces an affinity-based solid bioassay but also draws upon a deeper mechanistic understanding of target-ligand interactions, thereby enhancing the efficiency of CTC isolation.

## Methods
### Cell lines
MCF-7 (Cat. #: TCHu 74), HepG2 (Cat. #: TCHu 72) and A549 (Cat. #: TCHu150) cell lines were supplied by Stem Cell Bank, Chinese Academy of Sciences (China). MCF-10A (Cat. #: CL-0212), MDA-MB-231(Cat. #: CL-0150B), Ramos (Cat. #: CL-0483) and SK-Hep-1 (Cat. #: CL-0525) cell lines were obtained from Procell (China). All these cells were cultivated in DMEM culture medium (Gibco, USA) supplemented with 10% FBS (Gibco, USA) and 1% antibiotics (Gibco, USA) at 37 °C in 5% CO$_2$.

### Propagation and purification of M13 phage
M13 phage was propagated in Luria-Bertani (LB) medium using *E. coli* ER2738 (NEB, USA) as a host strain. The phages were first incubated with the bacteria culture at mid logarithmic growth phase for 5 h (MOI = 0.1, 37 °C, 250 rpm) and then separated from the host bacteria by centrifugation and concentrated by PEG/NaCl precipitation (20% (w/v) PEG-8000, 2.5 mol L$^{-1}$ NaCl) and suspended in Tris-HCl buffer (25 mmol L$^{-1}$ Tris-HCl, pH 7.5, 150 mmol L$^{-1}$ NaCl).

### Genetic modification of M13 phage
For the tip modification (pIII capsid protein), 6His tag was displayed to immobilize M13 phages on Ni-functionalized solid surface in an end-on way, due to its high chelating efficiency with Ni atom, moderate stability of this chelation, and the ease in operation[50–52]. In order to display 6His tag on the pIII protein, recombinant M13KE vector encoding 6His tag was constructed as we previously reported[24]. To be specific, two synthetic phosphorylated oligonucleotide strands 6H-Tag1/6H-Tag2 (Sequences listed in Supplementary Table 1, provided by Sangon Biotechnology, China) were hybridized by annealing at 95 °C for 5 min. M13KE vector (NEB, USA) was linearized by double-digestion with *Acc65I* and *EagI* (NEB, USA), followed by purification using gel purification kit (NEB, USA) and further ligation with the hybridization product (16 °C, 12 h). The recombinant M13KE was then transferred into chemical competent *E. coli* TG1 cells (Lucigen, USA) by CaCl$_2$ transformation method. The over-night-grown mono-clone on LB/IPTG/X-Gal agar plates were picked and validated by DNA sequencing.

### Hardening treatment of 6His-M13 phage
For evaluating the impact of flexibility of M13 phages on cell isolation performance, 6His-M13 phage was incubated with equal volume of 4% paraformaldehyde (PFA) solution (6 h, 4 °C, Dalian Meilun Biotechnology, China) and ultrafiltered for subsequent reactions, resulting in the formation of PFA-treated M13. Alternatively, 6His-M13 phage was also hardened by treatment with ethanol (100%, EtOH), generating EtOH-treated M13. The hardening treatment by EtOH was the same with that by PFA. Both PFA-treated M13 and EtOH-treated M13 were further subjected to the same procedure of azide modification and aptamer decoration. The amount of loaded aptamer on the PFA-treated rigid M13 was measured to be almost identical to that of the untreated flexible M13 (Supplementary Table 2).

### Azide modification of 6His-M13 phage
The 6His-M13 phages were chemically modified to incorporate azide groups by coupling with an active ester to form an amido bond in sterile PBS (pH 7.8) according to a modified protocol[53,54]. Specifically, the phages (10$^{11}$ pfu) were dissolved in 50 mM PBS (240 μL) with gentle mixing, followed by the addition of 250 μL N$_3$-PEG-NHS (10 mM, dissolved in DMSO, MeloPEG, China). After 18-h reaction (4 °C, 200 rpm), the excessive N$_3$-PEG-NHS were removed by ultrafiltration (MWCO = 100 kDa, Millipore, USA) and three-time washing by PBS. The resulting N$_3$-M13 were collected and stored at 4 °C for future use. Both N$_3$-M13 and unmodified M13 were desalted by ultrafiltration (MWCO = 100 kDa, Millipore, USA) and injected into Q-TOF LC/MS (Agilent Technologies, 1260-6540) for modification characterization.

### Aptamer-loading on N$_3$-M13 phage-immobilized magnetic microbeads (MBs)
In general, 1 mL of Ni-IDA functionalized magnetic microbeads (MBs, 2.5 mg mL$^{-1}$, Solarbio, China) were repeatedly washed with PBST (1×PBS), 3% BSA, tween-20 (0.01% (v/v), pH 7.4) followed by 3-h incubation with 5 × 10$^{10}$ pfu of N$_3$-M13 with gentle rotating at room temperature[50]. Subsequently, the N$_3$-M13 phage-anchored MBs (N$_3$-M13-MBs) were repeatedly rinsed with PBST and resuspended in 100 μL PBS for further aptamer decoration.

Dibenzocyclooctyne (DBCO)-functionalized aptamer S2.2[31] (DBCO-Apt, sequence listed in Supplementary Table 1, Sangon Biotechnology, China) was used for specific targeting of MUC1 positive cell lines. Aptamer decoration was achieved via click chemistry

between DBCO-Apt and N$_3$-M13. In brief, 100 µL of N$_3$-M13-MBs were allowed to react with 100 µL of DBCO-Apt (2.5 µM) for 6 h (200 rpm, 25 °C). The product of aptamer-decorated flexible M13-MBs (A-f-M13-MB) was repeatedly rinsed and resuspended in 500 µL PBS buffer. The counterpart MBs produced with rigid M13 phages was denoted as A-r-M13-MB for the comparison of cell isolation performance.

## Conjugation of aptamers on magnetic microbeads (MBs)

Aptamers were conjugated on magnetic microbeads (MBs) by a single-step carbodiimide reaction between NH$_2$-terminated aptamer and carboxyl MBs (Solarbio, China)[55]. Briefly, 500 µL of carboxyl MBs (2.5 mg mL$^{-1}$, suspended in 100 mM MES buffer, Sigma-Aldrich, USA) was activated by adding 250 µL EDC (50 mg mL$^{-1}$, Aladdin, China) and equal volume of NHS (20 mg mL$^{-1}$, Sangon Biotechnology, China), and incubated for 60 min at 25 °C on a rotator with 250 rpm shaking. The activated MBs were magnetically separated and repeatedly washed by PBS. Afterward, 24 µL of aptamer (10 µM) were incubated with MBs suspension overnight (25 °C). The product of aptamer-decorated MBs (A-MB) was obtained by magnetic separation and stored at 4 °C. The amount of aptamers decorated on MBs was estimated by decorating MBs with FAM-labeled aptamer and comparing the fluorescence difference between input FAM-aptamer-NH$_2$ and the supernatant after reaction.

## Aptamer-loading on N$_3$-M13

A total of 10$^{10}$ pfu of N$_3$-M13 were reacted with 1.25 µM (final concentration) DBCO-aptamer in PBS for 6 h (25 °C). Afterwards, the mixture was purified by PEG/NaCl precipitation six hours at 4 °C and 10-min centrifugation (9391 g) to obtain purified Apt-M13.

## Gel electrophoresis analysis and western blotting (WB) analysis

PAGE was conducted to characterize the assembly of Y-shaped DNA scaffold. A total of 10 µL DNA strand (1 µM) was added into 2.5 µL of 1 × loading buffer at 95 °C for 5 min. The electrophoresis was carried out with 12% polyacrylamide gel and run in 1 × Tris- Boric acid- EDTA running buffer (100 V, 60 min, 4 °C). The gels were further stained by 4 S Red Plus Nucleic Acid Stain (Sangon Biotechnology, China) and characterized by Tanon-4600SF automatic digital gel imaging analysis system (China).

For the evaluation of protein expression of cells before and after isolation and re-culture, CTC cells (MCF-7 as model cell, cell number: 5 × 10$^6$, Stem Cell Bank, Chinese Academy of Sciences) were lysed by RIPA buffer (Epizyme, China) for protein extraction. Afterwards, 20 µg of cellular protein (quantified by BCA protein detection kit, Epizyme, China) was loaded on 8–12% SDS-PAGE gel electrophoresis for separation. The proteins were transferred to 0.22 µm poly-vinylidene fluoride (PVDF, Cat. #: WJ001S, Epizyme, China) membrane and blocked with 5% skim milk at room temperature for 60 min to prevent non-specific binding. Then, the anti-ER antibody (Cat. #: ab32063, clone: E115, 1:500 dilution, Abcam, USA, https://www.abcam.cn/products/primary-antibodies/estrogen-receptor-alpha-antibody-e115-chip-grade-ab32063.html) and anti-GAPDH antibody (Cat. #: ab9485, clone: pAb, 1:2500 dilution, Abcam, USA, https://www.abcam.cn/products/primary-antibodies/gapdh-antibody-loading-control-ab9485.html) were diluted with 5% skim milk and separately added for overnight-incubation at 4 °C. The membrane was then washed six times with TBST (0.5% (v/v) tween 20) and incubated with the secondary antibody (HRP Conjugated AffiniPure Goat Anti-rabbit IgG (H + L) antibody, Cat. #: BA10541, clone: NA, 1:8000 dilution, Boster, China, http://boster.com/index/products/productsDetail?goods_sn=BA1054) at room temperature for 60 min. The membrane was washed with TBST and incubated with enhanced chemiluminescence (ECL) reagents (Epizyme, China) for chemilumi-nescence imaging (Bio Rad, USA). Quantitative analysis was conducted using ImageJ software.

## Young's modulus and stiffness assay of M13 phage

Briefly, about 10 µL M13 phage (10$^{10}$ pfu mL$^{-1}$) was pipetted onto the mica discs and let it stand for about 5 min, followed by the removal of excessive solvent with a filter paper. Thereafter, deionized (DI) water was pipetted three times to remove excessive salt and the remaining water was evaporated slowly at room temperature. Young's modulus and stiffness were measured under the contact mode using Bruker Dimension icon atomic force microscopy (AFM, Bruker, Dimension icon). The actual bending of the cantilever representing the deflection sensitivity and the spring contact of the cantilever measured in air was calibrated. Then, 5 points along a M13 phage nanofiber were selected to measure the Young's modulus and stiffness on at least ten samples and the data were averaged.

## Numerical simulations

Numerical simulations were conducted using COMSOL (version 5.6, COMSOL Ltd, USA) to simulate the force induced deformation of M13 nanofibers with different stiffnesses. The computing domain was set as 3 × 3 µm. M13 nanofiber was 880 nm in length ($Y$ direction), 6 nm in width ($X$ direction), and the elastic modulus $E$ was 0.5 GPa and 1.35 GPa for flexible and rigid M13, respectively. A M13 nanofiber was modeled as a linear elastic solid material. The direction of the flow field was set from the left boundary to the right with the inlet normal velocity given by Eq. (1). No-slip boundary conditions were imposed at all the other boundaries. Automatic remeshing was used to capture the dynamic motion of the M13 nanofibers.

$$\mathbf{u}_{in} = \frac{\mathbf{U} \times t^2}{\sqrt{(0.04 - t^2)^2 + (0.1t)^2}} \tag{1}$$

where $\mathbf{u}_{in}$ is the inlet normal velocity, $\mathbf{U}$ is the normal inflow speed, $t$ is the time.

## Determination of the binding affinity toward cancer cells

To assess the binding affinity of A-f-M13-MB toward MUC1 positive cells, 5 × 10$^4$ MCF-7 cells were first pre-stained with 4',6-diamidino-2'-phenylindole (DAPI, Boster, China) and then added into 500 µL of A-f-M13-MB suspension for 30 min-incubation at 37 °C. After incubation, the fluorescence intensity of the unbound cells in the supernatant was measured by a microplate reader (Synergy H1, BioTek, USA) at $\lambda_{Ex}/\lambda_{Em}$ = 360 nm/460 nm. Fluorescence from the captured cells was estimated by subtraction. By plotting the fluorescence intensity (FL) with the concentration of Apt ($C_{Apt}$), the binding affinity curve was obtained, which fitted the equation FL = $B_{max} C_{Apt}$ / ($K_d + C_{Apt}$), where $B_{max}$ refers to the maximum amount of binding sites, and $K_d$ refers to the binding constant, with a lower $K_d$ indicating a higher binding affinity. The $K_d$ of MCF-7 by A-r-M13-MB and A-MB was also measured using the same method to compare their CTC binding affinity. As for free apt, instead of collecting the fluorescence intensity of CTCs, fluorescence from FAM-aptamer was used and thus the $K_d$ value could be calculated similarly[47].

## Preparation of Ni-IDA grafted glass slides for M13 phage grafting

The Ni IDA-grafted glass slides were prepared according to our previous work[56]. Firstly, the glass slides (1 × 1 cm$^2$) were first treated with piranha solution to activate hydroxyl groups, and then placed in 5 mL tubes and immersed in alkaline solution for 20 min. They were then immersed into 170 µL epichlorohydrin (Tianjin Damao Chemical Reagent, China) under ultrasonic bath for 30 min for epoxy-functionalization. Next, the epoxy glass slides were reacted with imi-nodiacetic acid (Sinopharm, China) at 70 °C for 10 h and further che-lated with nickel ion (0.1 M) at 25 °C for 3 h. The glass slides were carefully rinsed with DI water between every two steps. The resulting Ni-IDA grafted glass slides could anchor 6His-M13 or N$_3$-M13 through

6His-tagged pIII protein in an end-on manner. After grafting of $N_3$-M13 ($10^{10}$ pfu) on Ni-IDA grafted glass slide, DBCO-Apt was clicked on the slide with procedure identical to "Methods" section "Aptamer-loading on $N_3$-M13 phage-immobilized magnetic microbeads (MBs)", resulting in A-f-M13-slide. The counterpart slides grafted with rigid M13 phages was denoted as A-r-M13-slide for the comparison of cell adhesion force. For the preparation of the aptamer-decorated slide, FAM-aptamer-$NH_2$ was used to conjugate with the epoxy glass slides after 10 h-reaction at 25 °C, resulting in A-slide.

## Cell adhesion force measurements

The adhesion force between cell population and M13 nanofibers was evaluated by a reversed centrifugation force method according to Jia's work[38]. Briefly, $10^4$ MCF-7 cells (or $10^6$ Ramos cells, Procell, China) were pre-stained with acridine orange (AO, Beyotime, China) and incubated with A-f-M13-slide in a 24-well plate for 30 min. Uncaptured cells were gently washed off from the slide with PBS buffer, and remaining cells on the slide was counted under the fluorescence microscope (10×objective). Next, the slide was inverted and placed on the bottom of a 15 mL centrifuge tube that was filled with 10 mL PBS and centrifuged with centrifugal forces ranging from 33 to 1905 × g for 5 min. After that, the number of the remaining cells on the slides was enumerated under fluorescence microscope. The percentage of the remaining cell on the slide at different centrifugal force was calculated as follows:

$$\text{Remaining cells\%} = \frac{\text{Number of remained cell after centrifugation}}{\text{Number of cell before centrifugation}} \times 100\%$$
(2)

The cell adhesion force was equal to the relative centrifugal force when 50% cells remained on the slide. The relative centrifugal force of cells can be calculated using the following equation:

$$\mathbf{F}_c = (\rho_{cell} - \rho_{medium}) \times V_{cell} \times \boldsymbol{\omega}^2 \times (r_0 + x)$$
(3)

where $\mathbf{F}_c$ is the relative centrifugal force, $\rho$ is the density of cells or 1 × PBS medium (the cell density is about 1.07 g cm$^{-3}$ and the PBS medium density is about 1 g cm$^{-3}$), $V_{cell}$ is the volume of cell (4000 μm³ for MCF-7 cells and 500 μm³ for Ramos cells), $\boldsymbol{\omega}$ and $r_O$ are the angular speed and radius of centrifugation, $x$ is the lateral distance from the bottom of the tube to the center of centrifuge. In order to get a fair comparison, the amount of loaded aptamers was kept the same for A-f-M13-slide, A-r-M13-slide and A-slide.

## Molecular modeling

Dissipative particle dynamics (DPD) simulations were used to better understand the interactions in the fluid between the receptors on CTCs and the aptamers on MB and A-f/r-M13-MB systems at the molecular level. Due to its large size, the microbead (MB) was modeled as a planar substrate in the simulations. As a result, the A-MB was modeled as a planar substrate with the aptamer covalently decorating the substrate; the aptamer was modeled as a linear polymer made of beads with one bead at the end that can interact with the specific beads constituting the receptor. The total number of the aptamers in A-MB was 196. The A-f-M13-MB was modeled as a planar substrate with sixteen cylinders (with each cylinder representing a M13 phage) coated on its surface. The neighboring beads in the cylinder were connected by a harmonic bond to ensure its integrality, allowing the cylinder to freely twist or bend in the simulations. Twelve aptamers were covalently decorated on each cylinder, reaching 192 aptamers in A-f-M13-MB, similar to A-MB. The A-r-M13-MB was the same as the A-f-M13-MB except that the cylinder was treated as a rigid body, thus the cylinder was unable to freely twist or bend in the simulations. A WBC was modeled as a lipid vesicle, which was fabricated by self-assembling of about 2000 lipid

molecules. A CTC was modeled as a lipid vesicle with some receptors on its surface, where the lipid was the same as that in the WBC and the receptor was modeled as the branched polymer with one specific bead at the terminal of each branch.

DPD is a coarse-grained (CG) simulation technique with hydrodynamic interaction[39]. The dynamics of the beads are governed by Newton's equation of motion. The total force exerted on each bead typically includes three parts: $\mathbf{F}_i = \sum_{i \neq j}(\mathbf{F}_{ij}^C + \mathbf{F}_{ij}^D + \mathbf{F}_{ij}^R)$ where $\mathbf{F}_{ij}^C$ is the conservative force; $\mathbf{F}_{ij}^D$ is the dissipative force; and $\mathbf{F}_{ij}^R$ is the random force. To denote the hydrophilic/hydrophobic property of the beads, we set the repulsive parameter $a_{ij}$ in conservative force as 25 $k_BT/r_c$ if $i = j$, and $a_{ij} = 100$ $k_BT/r_c$ if $i \neq j$[57,58]. Moreover, we used the "soft" Lennard-Jones (LJ) potential[59] to model the strong receptor-aptamer specific interaction and the weak lipid head-aptamer non-specific interaction. The interaction strength was set as 6.0 $k_BT$ in the former case and it was set as 1.5 $k_BT$ in the latter case. In addition, the harmonic spring interaction $U_S = k_s(r_{i,i+1} - l_0)^2$ between neighboring beads in a single molecule was used to ensure the integrality of lipids, aptamers and M13[57,58], where $k_s$ is the spring constant and $l_O$ is the equilibrium length and the parameter $k_s = 64$ $k_BT$, $l_O = 0.5$ $r_c$ was used. We further used a three-body angle potential $U_a = k_a(1 - \cos(\phi - \phi_0))$ to depict the rigidity of lipid tails and aptamer ($k_a = 10.0$ $k_BT$ and $\phi_0 = 180°$)[57,58], where $k_a$ is the energy constant and $\phi_0$ is the equilibrium value of the angle.

The velocity-Verlet integration algorithm was used to integrate Newton's equations of motion with the integration time step of 0.02 $\tau$. For the sake of simplicity, we chose the bead mass $m$, the cutoff radius $r_c$, and the energy $k_BT$ as the reduced units in the simulations. The simulations were performed in the NVT ensembles with the periodic boundary conditions adopted in the three directions. The size of the box was set as 60×60×50 and the number density of the beads was set as 3.0 in the simulations. All the simulations were carried out using the modified software package Lammps (12 Dec 2018)[60].

## Cell capture

The A-f-M13-MB was pre-blocked with a blocking solution (3% BSA in PBS) to reduce non-specific adsorption caused by bare MB surfaces before cell capture experiments. Firstly, both the positive or negative cells were harvested using 0.25% trypsin and immediately resuspended in PBS buffer. Next, after enumeration with a hemocytometer, cell suspensions were diluted to the desired concentrations before using. Thereafter, the model cells were either resuspended in the PBS buffer, or stained with AO and resuspended in the whole blood samples. Afterwards, the cell suspension (500 μL) was incubated with equal volume of A-f-M13-MB for 30 min at 37 °C. Subsequently, the uncaptured cells was magnetically separated and counted under fluorescence microscope (Olympus, BX53M)[61].

In order to evaluate the capture performance of A-f-M13-MB in a real sample matrix, positive cells with a definite number were added to the freshly collected whole blood obtained from healthy volunteers, and the spiked sample was used to study the cell capture by A-f-M13-MB. The cell capture procedure was the same as described above, with the cell capture efficiency calculated using equation (4). All experiments were performed three times.

$$\text{Cell captured\%}$$
$$= \frac{\text{Total Number of added cells} - \text{Number of uncaptured cells}}{\text{Number of input cells}} \times 100\%$$
(4)

## Release assay, cell viability analyses and cell migration activity analyses

For the cell release assay, DNase I (Takara, China) was used to destroy the phosphodiester bond within the aptamers to release the captured cells. In this work, to determine the optimal release conditions, the

dependance of the cell release efficiency on the concentration and incubation time of DNase I were investigated. Under optimized conditions, the captured cells from the buffer or the blood by A-f-M13-MB were magnetically separated, suspended in PBS buffer and incubated for 20 min with 100 U mL$^{-1}$ DNase I at 37 °C. Then, the cells in the supernatant were collected and enumerated. The cell release efficiency was derived using the following equation:

$$Cell\ released\% = \frac{Number\ of\ released\ cells}{Number\ of\ captured\ cells} \times 100\% \qquad (5)$$

Cell viability was evaluated by acridine orange (AO)/ propidium iodide (PI) assay (ApexBio, USA) according to the operating manual. The cells with high viability were stained by AO to emit a green fluorescence, whereas these dead cells emitted red fluorescence due to the labeling by PI. By counting the red emitting cells and green emitting cells, the viability can be calculated. To further validate the viability of released cells, they were cultured at 37 °C, 5% $CO_2$ with DMEM cell culture medium.

For evaluating the migration activity of CTCs before and after isolation and re-culture, 6 mL of cell suspension ($5 \times 10^4$ cells mL$^{-1}$) was incubated in a 25 cm$^2$ culture flask for 3 h for cell adhesion. The Live cell Station (CytoSMART Lux3, Axion BioSystems, USA) were used to collet cell migration images at a fixed area every 5 min for 4 h. Subsequently, ImageJ (version 1.53t) was used for the analysis of the average moving distance and speed and the movement trajectory of cells was obtained using Matlab software (version R2020b).

## CTC purity during capture and release processes

We took the Ramos cells as white blood cell (WBC) model to investigate the CTC purity during capture and release. WBCs were stained by DAPI and MCF-7 cells were stained by AO for discrimination. The cells were counted by a hemocytometer. Approximately 100–1000 MCF-7 cells were mixed evenly with $10^6$ WBCs. Afterwards, A-f-M13-MB was mixed with the cell mixture for 30 min-incubation at 37 °C to capture target CTCs. Then, the uncaptured cells were collected and counted under fluorescence microscope. The captured cells were subsequently released using DNase I and counted. The number of nonspecifically attached WBCs were obtained from the number of initial input WBCs subtracting the number of uncaptured WBCs. The purity of CTC during capture and release processes was evaluated by depletion index value. The WBCs depletion index was calculated according to the following equation:

$$WBCs\ depletion = \log_{10}\left(\frac{Number\ of\ initial\ WBCs}{Number\ of\ final\ WBCs}\right) \qquad (6)$$

## Study design

For evaluating the diagnostic efficacy of A-f-M13-MB, a total of 100 participants were enrolled in this study, including 56 BC patients, 34 benign patients, and 10 healthy volunteers. As breast cancer is mostly suffered by female patients, most of the participants involved in this study were female (95%). Considering the possibility of male patients that suffer from benign breast disease, the cohort involved four healthy male donors and one male benign patient (See detail information in the Excel file "Supplementary Data 1. Information of clinical specimen"). All participants were recruited from Liaoning Cancer Hospital. Only patients with definite information of sex, age, and pathological diagnosis were recruited. The study complied with all relevant ethical regulations and was approved by the Ethics Committee of both Northeastern University, China (No. NEU-EC-2021B020S) and Liaoning Cancer Hospital (20211035). All individuals were anonymized, and only gender, age, pathological diagnosis, treatment plan and treatment response were recorded. The requirement for consent was waived by the ethical review body, as we collected remnant samples destined for disposal on the day of sample request from the study of the co-author (J.Y.L.). The consent to publish the participants' information regarding their age and sex were obtained by the co-author (J.Y.L). No self-selection criteria bias for patient populations was anticipated.

## Isolation and analysis of CTCs from breast cancer patients

500 μL of breast cancer patient's whole blood without any treatment was incubated with 500 μL A-f-M13-MB at 37 °C for 30 min under gently shaking. After magnetic separation, the captured CTCs were released using 100 U mL$^{-1}$ DNase I at 37 °C for 20 min. Finally, the released cells were identified using CTC immunofluorescence staining kit (IFH-001, Cytelligen, USA). Briefly, the released cells were fixed with 4.0% formaldehyde at 33 °C overnight. They were then blocked with 3% goat serum followed by incubation with antibody cocktail (EpCAM antibody conjugated with AF488, anti-CK conjugated with Cy5 and anti-CD45 conjugated with AF594) and DAPI for observation using a fluorescent microscope. The isolated CTCs in breast cancer patients' blood were identified as DAPI$^+$/EpCAM$^+$/CK$^+$/CD45$^-$ cells, whereas WBCs were identified as DAPI$^+$/EpCAM$^-$/CK$^-$/CD45$^+$ cells. Meanwhile, to access the molecular profiling of the captured cells, the anti-HER2 antibody (Cat. #: ab237060, clone: EP2324Y, 1:100 dilution, Abcam, USA https://www.abcam.cn/products/primary-antibodies/alexa-fluor-488-erbb2--her2-phospho-y877-antibody-ep2324y-ab237060.html), anti-estrogen receptor alpha antibody (Cat. #: ab205851, Clone: EPR4097, 1:50 dilution, Abcam, USA, https://www.abcam.cn/products/primary-antibodies/alexa-fluor-647-estrogen-receptor-alpha-antibody-epr4097-ab205851.html), anti-CD45 antibody and DAPI were used according the same staining process. The luminal cells were identified as DAPI$^+$/ER$^+$/HER2$^{-or+}$/CD45$^-$ cells, HER2 cells were identified as DAPI$^+$/ER$^-$/HER2$^+$/CD45$^-$ cells, and triple-negative cells were identified as DAPI$^+$/ER$^-$/HER2$^-$/CD45$^-$ cells.

## Isolation of CTC mixtures with different EMT subphenotypes by A-f-M13-MB bearing Y-shaped DNA scaffold

The Y-shaped DNA scaffold was assembled by annealing $Y_1$, $Y_2$ and $Y_3$ (1.25 μM for each strand) at 95 °C for 6 min, and attached to 2.5 mg mL$^{-1}$ 1 mL of M13-MBs via click reaction following the same procedure described in Methods section titled "Aptamer-loading on $N_3$-M13 phage-immobilized magnetic microbeads (MBs)". The sequences of $Y_1$, $Y_2$ and $Y_3$ are listed in Supplementary Table 1 (Sangon Biotechnology, China). For evaluating the CTC isolation performance of the resultant A-f-M13-MB, CTC mixtures (contain 1000 CTCs in total) were prepared by spiking MCF-7 cell (E-type CTC model) and MDA-MB-231 cell (M-type CTC model) in CTC-free whole blood at a molar ratio of 9:1, 3:1, 1:1, 1:3 and 1:9. The CTC mixtures were then captured following procedures described in Methods section titled "Isolation and analysis of CTCs from breast cancer patients", except for the additional immunofluorescence staining by Anti-E Cadherin antibody (Cat. #: ab40772, clone: EP700Y, 1:1000 dilution, Abcam, USA, https://www.abcam.cn/products/primary-antibodies/e-cadherin-antibody-ep700y-intercellular-junction-marker-ab40772.html) and Goat Anti-Rabbit IgG H&L (Alexa Fluor® 488) antibody (Cat. #: ab150077, clone: NA, 1:200 dilution, Abcam, USA, https://www.abcam.cn/products/secondary-antibodies/goat-rabbit-igg-hl-alexa-fluor-488-ab150077.html), Anti-N Cadherin antibody (Cat. #: ab98952, clone: 5D5, 1:200 dilution, Abcam, USA, https://www.abcam.cn/products/primary-antibodies/n-cadherin-antibody-5d5-intercellular-junction-marker-ab98952.html), and Goat Anti-Mouse IgG H&L (Alexa Fluor® 594) (Cat. #: ab150116, clone: NA, 1:200 dilution, Abcam, USA, https://www.abcam.cn/products/secondary-antibodies/goat-mouse-igg-hl-alexa-fluor-594-ab150116.html).

## Statistics and reproducibility

IBM SPSS Statistics software (version 19.0) was used to test the difference for diagnosis stages from different CTCs using a two-tailed *t*

test. The differences were considered significant at $p < 0.05$ (**$p < 0.01$, ***$p < 0.001$, ****$p < 0.0001$), and considered to have no significance at $p > 0.05$ (ns). Confusion matrix for subtype diagnosis by CTC molecular profiling method was drawn by Origin software (version 2021). Receiver operating characteristic (ROC) curve was plotted using the MedCalc statistical software (version 20. 010), presenting the diagnostic accuracy. The results of dissipative particle dynamics simulations were analyzed by Lammps (version 12 Dec 2018) and GraphPad prism (version 9.3.0). The mean distance and migration speed of cells before and after isolation and re-culture were collected using Image J software (version 1.53t) with Manual Tracking plugin. The movement trajectory of cells before and after isolation and re-culture was analyzed using Matlab software (version R2020b). All experiments were repeated independently with similar results for at least three times, especially micrograph results.

## Reporting summary

Further information on research design is available in the Nature Portfolio Reporting Summary linked to this article.

## Data availability

All data are presented in the main text and the supplementary files. Source data are provided with this paper. The raw data used in this study are available in the Figshare database (https://figshare.com/s/9e44959bd453d88a55f5) under accession DOI code 10.6084/m9.figshare.25757205. Source data are provided with this paper.

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

## Acknowledgements

T.Y. would like to acknowledge the financial support from the Natural Science Foundation of China (22374014, 22174011), the Fundamental Research Funds for the Central Universities (N2305018) and the Liaoning Revitalization Talents Program (XLYC2007102). H.M.D. would like to acknowledge the financial support from the Natural Science Foundation of China (12222506, 12347102). J.H.W. would like to acknowledge the financial support from the Natural Science Foundation of China (22074011). C.M. would also like to acknowledge the financial support from the General Research Fund of Hong Kong (14208723). Special thanks are due to the instrumental analysis from Analytical and Testing Center, Northeastern University. The authors would also like to thank Tingli Luo and Yicai Jia at Cancer Hospital of China Medical University for their contribution in the cell migration and phenotype drifting experiments.

## Author contributions

T.Y., J.H.W. and C.M. supervised the project and wrote the manuscript. T.Y. and C.M. conceived the research and designed the experiments. H.D.L. carried out most of the experiments, including the construction and characterization of the capture surface, the measurement of the cell binding affinity and adhesion force, the evaluation of CTC isolation performance of the capture surface, and the analysis of clinical samples, under the supervision of T.Y. Y.Q.C. carried out the dissipative particle dynamics (DPD) simulation experiments under the supervision of H.M.D. J.Y.L. (Jing-Yue Li) and C. W. collected the clinical samples and made the clinical subtype diagnosis reports under the supervision of J.Y.L. (Jian-Yi Li). X.W. carried out the numerical simulation experiment. S.Y.W. assisted in the affinity calculation, and Y.C. provided necessary help in the collection of the TEM images. R.W. and Y.L. participated in the design of the illustrations. All the authors contributed to the discussions and writing of the manuscript.

## Competing interests

The authors declare no competing interests.
