## [Peer Review File · Nature Communications]

REVIEWER COMMENTS

Reviewer #1 (Remarks to the Author):

This manuscript reports a strategy for the capture of circulating tumor cells based on surface scaffolds, either glass slides or magnetic beads, that are modified with M13 phages. To ionically attach the phages to such surfaces, the phages were genetically engineered to produce a variant capsid protein displaying six his tags, and the surfaces modified with Ni – iminodiacetic acid. Additionally, the phages were further modified via coupling reactions between N-terminal amines of lateral proteins pVIII and the linker NHS-PEG-N3. This linker was next coupled to nucleic acid aptamers via click chemistry on the azide. The goal was to use the phages as flexible, multivalent receptors, relying on the large number of aptamers attached to the phage sides for molecular recognition of tumor cells. The aptamers employed in this work were selected against Mucin 1 (MUC1); the aptamer sequences were selected by a different study provided in ref. 30 (I could not assess the aptamer performance due to lack of access to the manuscript behind the paywall). The manuscript claims three main results: (1) an improvement in apparent affinity for MUC1 positive cells relative to PFA-treated phages, magnetic beads, or the aptamers in solution, (2) a reduction in non-specific binding by white blood cells, and (3) improved capture and release of MUC1 positive cells relative to PFA-treated phages and aptamers on beads. Overall, the approach is interesting because it relies on a biological scaffold, the phages, to achieve sensitive cell capture with decreased non-specific fouling. I believe this work would be of interest to the broad readership of this journal. However, in my opinion, the data presented does not rigorously support some of the claims made by the manuscript, the narrative lacks key experimental information that is necessary to improve understanding of the results presented, and some of the figures require additional labeling and explanations. My major concerns are indicated below:

1.- The assay used to measure binding of the phages to MUC1 positive cells is not conventional and should be briefly described in the main text (it is currently deferred to the SI). Binding of the phages to the cells is measured differentially. Regardless of the equation used to fit the data, these measurements are not directly reporting molecular binding and, therefore, Kds. Instead, I suggest the use of EC50s as nomenclature, reflecting the half maximal cell concentration measurable in this assay format.

2.- The EC50 of A-f-M13-MB (flexible) appears to be one order of magnitude better than the EC50 for A-r-M13-MB (rigid). This is unsurprising given that treatment of the aptamers with PFA can result in modification of the nucleotides. This experiment does not conclusively demonstrate that flexible is better than rigid in terms of affinity. Additionally, PFA treatment should result in the modification of phage proteins, also potentially affecting binding in ways not related to "rigidity".

3.- Fig 4a should show the dose-response curves on the same x axis scale. The current differences in axes artificially enhance the sharpness of the curve in the first panel relative to the others.

4.- In Fig.2a, it is not clear what the TEM micrograph is intending to show. The image should be annotated to highlight the features relevant to the discussion.

5.- In Fig. 2e, the gel as presented is meaningless. The molecular size marker should be labeled. The band for WT is smudged and seems to reflect a huge excess of sample in the run and/or lack of purity. The full lanes must be shown with proper labeling. Any additional bands appearing in the gel should be explained. The gel is also very blurred and unnecessarily small in the current version.

6.- In Fig. 3c-e, the loading of the M13 phages under the various conditions should be included in the caption. It is not clear if the differences between AFM panels are simply due to differences in loading of phages onto the substrates.

7.- Also in Fig. 3, the simulations are not rigorously discussed in the manuscript, and they are not matched to any relevant experimental measurements. My suggestion is to remove them since they are not given proper attention in the work and the assumptions made to create the models are not transparently discussed in the main text. These data also seem redundant with the data in Fig. 3a,b.

8.- It is nice to think that the modification of the phages with many aptamers should increase the valency of the interaction. However, there are no measurements in the manuscript directly demonstrating multivalency. This claim should be removed or direct evidence provided that multiple aptamers are able to bind to each single cancer cell thanks to the added flexibility. Given the chemical effect that PFA may have on the nucleotides and phage proteins, the "rigid" system is not a good reference control for a claim of mono- vs -multivalency. For example, there could be crosslinking happening in the PFA-treated phages that prevents the same extent of interaction as the untreated phages, regardless of rigidity.

9.- On page 11, the manuscript reads: "the numerous aptamers on M13 nanofibers have the capacity to replenish any degraded aptamers nearby, and they can dynamically adjust their positions by freely twisting the M13 scaffold". This is a nice concept; however, there is no rigorous experimental evidence of any of these claims in the manuscript.

Reviewer #2 (Remarks to the Author):

In this work by Hui-Da Li et al, M13 phage is genetically modified to be used for isolation and count of CTCs from the blood from breast cancer patients, in the format of nanofibers. This can be used for efficient diagnosis of breast cancer subtypes.

Even though this work has novelty and is well executed, the overall idea and the field around specifically M13 phage to be used as biomaterial or relevant for diagnostics or other applications is not new. This recent review is just an example capturing the progress in the field (<https://doi.org/10.1016/j.mtbio.2023.100612>).

The experiments are very carefully executed overall, the data are sound, but it is not clear how stable this genetically modified phage is over time. Is stable enough to be able to be used as a diagnostic regularly or is this a one-off study with modified phage freshly prepared for each experiment? Usually modified phages are not very stable. Also, how was the specific modification selected-what were the specific criteria that these sequences were selected for this phage modification-please clarify on the methods and the text.

I understand why M13 was selected as an example phage, but are there any other potential phages that could work equally well with this methodology?

It is also a bit confusing what is the overall aim of the paper. Is it to provide a new diagnostic tool for cancer or is it to provide a new methodology that relates to M13 mechanical properties? It has to be phrased in a way that is clear and potentially more clinical samples can be tested-also from other cancers and not only breast cancer.

The methodology otherwise is sound and the methods are well written and reproducible.

Some further comments:figure 6 (e) is not very clear-please demonstrate the result clearly. figure 7 (d) some text on the side is cut-please amend.

supplementary figure 4 (a-c) (d-f)-f is missing from parenthesis, please amend and increase contrast so that is obvious what you want to demonstrate.

Reviewer #3 (Remarks to the Author):

In this manuscript, a versatile phage nanofiber system was utilized to enhance the capture efficiency of CTCs and inhibit the non-specific adsorption of WBCs by modifying the high peak degree of aptamer on the nanostructures of this fiber to increase the binding affinity between the

ligands on the functional microspheres and the markers of the circulating tumor cell membranes with the help of the nanofiber's flexibility and deformability. This approach is very interesting and the newly-developed probes exhibit an overwhelming strength in CTCs capture over the previously reported nanoworms-like probes due to much more flexible chain of phage M13. The authors provide adequate experimental data to testify the feasibility of the probes in highly efficient enrichment of CTCs based on the samples with spiked tumor cells and clinical patients samples. However, to be one of the valuable articles in the research field of CTCs capture, the following key issues should be considered in order to attract more readers, because some microfluidic microarray-based systems have already been able to realize the capture of high-purity and high-sensitivity CTCs for prognostic monitoring applications. Additionally, as for rapid culture and proliferation of CTCs in vitro after isolation from clinical patients' blood samples for precise diagnosis and early warning related to metastatic recurrence of cancer patients, or the use of proliferated CTCs cell spheres to construct a mouse transplantation xenograft to study new intervention methods of tumor metastasis, all of these require that the release and detachment of CTCs from isolation probes can be achieved in a highly active and efficient manner. Although with some advantages in improving purity and capture efficiency of CTCs compared to ordinary magnetic probes, this probe still needs to be improved in terms of the downstream applications requirements. So, this manuscript can't be accepted for publication in this journal before following concerns and questions have been well addressed.

1.The quality of the figures need to be carefully checked and improved, such as several panel numbers are missing in Figure 5; the curves in panel e are not labeled, and the vertical axes are not labeled with physical quantities. Similar errors happened to Figure 3a, Figure 6 and 7.

2.As a powerful tool of CTCs enrichment, it is also necessary to consider to carry out the high availability release, the special structure and reactivity of M13 phage as well as the flexible characteristics contributed to higher binding affinity of CTCs, how to realize such CTCs release from the probe after CTCs enrichment upon the M13 phage based system?

3.It is recommended that the curve in Figure 7c be replaced with an AUC curve and the corresponding table of raw data be provided.

4.The authors claimed that “ An average of approximately 622 aptamers was found on the sidewall of each Apt-M13 (Supplementary Fig. 3), how did they quantify the M13?

5.As we know, CTCs have different EMT subtypes, and a large number of aptamer molecules are able to be attached to this M13 phase nanofibers, it is strongly suggested to examine whether it is possible to isolate CTC subphenotypes of cells covering different EMT characteristics from the clinical patients' blood samples if several ligands targeting E-type and M-type CTCs were attached to a single phage chain at the same time?

Point-by-Point Responses to Reviewers

Reviewer 1:

General Comments:

This manuscript reports a strategy for the capture of circulating tumor cells based on surface scaffolds, either glass slides or magnetic beads, that are modified with M13 phages. To ionically attach the phages to such surfaces, the phages were genetically engineered to produce a variant capsid protein displaying six his tags, and the surfaces modified with Ni – iminodiacetic acid. Additionally, the phages were further modified via coupling reactions between N-terminal amines of lateral proteins pVIII and the linker NHS-PEG-N₃. This linker was next coupled to nucleic acid aptamers via click chemistry on the azide. The goal was to use the phages as flexible, multivalent receptors, relying on the large number of aptamers attached to the phage sides for molecular recognition of tumor cells. The aptamers employed in this work were selected against Mucin 1 (MUC1); the aptamer sequences were selected by a different study provided in ref. 30 (I could not assess the aptamer performance due to lack of access to the manuscript behind the paywall). The manuscript claims three main results: (1) an improvement in apparent affinity for MUC1 positive cells relative to PFA-treated phages, magnetic beads, or the aptamers in solution, (2) a reduction in non-specific binding by white blood cells, and (3) improved capture and release of MUC1 positive cells relative to PFA-treated phages and aptamers on beads. **Overall, the approach is interesting because it relies on a biological scaffold, the phages, to achieve sensitive cell capture with decreased non-specific fouling. I believe this work would be of interest to the broad readership of this journal.** However, in my opinion, the data presented does not rigorously support some of the claims made by the manuscript, the narrative lacks key experimental information that is necessary to improve understanding of the results presented, and some of the figures require additional labeling and explanations.

Response:

Reviewer 1 specific comment:

Q1. The assay used to measure binding of the phages to MUC1 positive cells is not conventional and should be briefly described in the main text (it is currently deferred to the SI). Binding of the phages to the cells is measured differentially. Regardless of

the equation used to fit the data, these measurements are not directly reporting molecular binding and, therefore, K_d s. Instead, I suggest the use of EC_{50} s as nomenclature, reflecting the half maximal cell concentration measurable in this assay format.

Answer: We appreciate the reviewer for the constructive comment. In this study, we mainly adopted **three ways** to evaluate the binding between phage-based solid surface and the target cell (MUC-positive cell), namely, binding constant (dissociation constant, K_d), cell adhesion force, and the total energy of receptor-aptamer interaction. While K_d estimates how tightly the ligand (phage) is bound to the receptor (target cell) at the equilibrium state, it actually cannot directly report the intrinsic nature of molecular binding. We therefore employed the cell adhesion force as a complementary property to study the binding **at the cell-affinity interface level (microscopic level)**, which reflects the whole contribution of multiple ligands on the same solid surface. Additionally, the total energy of receptor-aptamer interaction was estimated by theoretical calculation, which provides deeper insight **at the aptamer-MUC1 level (molecular level)**. As EC_{50} is derived similarly as K_d , it also only provides binding information at **the macroscopic level (solution level)**. We therefore prefer to keep the use of the three methods in our manuscript as they provide molecular binding information from three different perspectives.

Q2. The EC_{50} of A-f-M13-MB (flexible) appears to be one order of magnitude better than the EC_{50} for A-r-M13-MB (rigid). This is unsurprising given that treatment of the aptamers with PFA can result in modification of the nucleotides. This experiment does not conclusively demonstrate that flexible is better than rigid in terms of affinity. Additionally, PFA treatment should result in the modification of phage proteins, also potentially affecting binding in ways not related to "rigidity".

Answer: We appreciate the reviewer for the constructive comment. In this study, aptamers are mainly responsible for the binding to CTC cells, and the phage nanofiber mainly acts as a scaffold to enable better performance of the aptamer. **The treatment would not result in the modification of the nucleotides (aptamers), because we**

treated the phages with PFA first and then conjugated them with aptamer. Therefore, the PFA treatment will not affect the aptamer binding affinity and the decreased affinity we observed was not due to the deactivation of aptamers after treatment but due to the PFA-induced rigidity of the phage.

The PFA treatment would result in modification of phage proteins, as the carbonyl group ($-C=O$) in PFA molecule would react with the primary amine group ($-NH_2$) of phage capsid proteins to form Schiff base product (contains $-C=N-$ bond). The mutual crosslinking of phage capsid proteins mediated by PFA thereby results in the increased rigidity. Nevertheless, **after PFA treatment, sufficient reactive functional groups remained on the phage surface and were available for aptamer conjugation, and thus the aptamer loading on the PFA-treated phage was found to be more than that on the untreated phage** (Supplementary Table 2, untreated phage: 622 aptamers on each phage, PFA-treated phage: 756 aptamers on each phage). **Under such a circumstance, the A-f-M13-MB (flexible) still outperformed A-r-M13-MB (rigid) in terms of affinity (Fig. 4a).** Therefore, we came to the conclusion that change in rigidity was mainly responsible for the difference in affinity.

We have to admit that there's no ideal way to change the rigidity without changing other properties. Therefore, we employed dissipative particle dynamics (DPD) simulations to explore the influence of rigidity on the interactions between CTCs and MBs. In the simulation, flexible phage was modeled as a cylinder. The neighboring beads in the cylinder were connected by a harmonic bond to ensure its integrality, allowing the cylinder to freely twist or bend in the simulations. In the case for rigid phage, the cylinder was treated as a rigid body, thus the cylinder was unable to freely twist or bend in the simulations. **As indicated in the simulation results, the rigid phage exhibits a lower relative density of aptamers (positively correlated to RDF value) around the target receptors than the flexible phage (Figure 4f).** As more contacts between the specific beads constituting the aptamers and the receptors resulted in lower total energy of the receptor-aptamer interaction, A-f-M13-MB (flexible) has lower total energy than A-r-M13-MB (rigid) when interacting with CTCs (Figure 4g). These results theoretically demonstrated the influence of rigidity/flexibility

on the interaction between CTCs and MBs.

Having said these, we have revised the original description regarding the preparation of rigid counterparts (Section S2. Aptamer loading amount on M13 with different stiffnesses in the SI file, and Section “Mechanical properties and dynamic movement of the engineered M13 phage” in the main text.) **to improve the understanding of the results presented as the reviewer suggested.**

Q3. Fig 4a should show the dose-response curves on the same x axis scale. The current differences in axes artificially enhance the sharpness of the curve in the first panel relative to the others.

Answer: We appreciate the reviewer’s suggestion. We adjust the x axis scale of the first two panels in Figure 4a to be the same as the reviewer suggested (See Figure R1). The x axis scale of the last two panels is in the range of 0-800 nM, which is over three magnitudes larger than that of the first two panels. If we adjust the x axis scale of the first two panels to be 0-800 nM, the curves in the first two panels would be even sharper. Therefore, it is not appropriate to adjust all the four x axis scales to be the same.

Figure R1. The dissociation constants of (a) A-f-M13-MB and (b) A-r-M13-MB derived from their corresponding affinity curve. (Added to revised manuscript as part of Figure 4a)

Q4. In Fig.2a, it is not clear what the TEM micrograph is intending to show. The image should be annotated to highlight the features relevant to the discussion.

Answer: We appreciate the reviewer for the thoughtful suggestion. We have added arrows in the TEM micrograph to indicate the location of M13 nanofibers as the reviewer suggested in the revised manuscript (See Figure R2).

Figure R2. TEM images of A-f-M13-MB. The arrows indicated that M13 nanofibers were anchored on MBs in an end-on manner. Scale bar: 500 nm. (Added to revised manuscript as part of Figure 2a)

Q5. In Fig. 2e, the gel as presented is meaningless. The molecular size marker should be labeled. The band for WT is smudged and seems to reflect a huge excess of sample in the run and/or lack of purity. The full lanes must be shown with proper labeling. Any additional bands appearing in the gel should be explained. The gel is also very blurred and unnecessarily small in the current version.

Answer: We apologize for not providing high quality gel results. Considering evidence from the fluorescence microscopic images (Fig. 2) has already well demonstrated the successful decoration of aptamers on phage, where the MBs anchored with A-f-M13 emitted green fluorescence while the control MBs anchored with WT-M13 didn't, we decided to delete the low-quality gel results in the revised manuscript.

Q6. In Fig. 3c-e, the loading of the M13 phages under the various conditions should be included in the caption. It is not clear if the differences between AFM panels are simply due to differences in loading of phages onto the substrates.

Answer: We appreciate the reviewer for the thoughtful suggestion. We agree with the reviewer that the loading of M13 phages in Figure 3c-e should be the same to guarantee the differences between AFM panels are not originated from the loading

differences. Therefore, we carried out this experiment again and controlled the loading of the three types of phages to be the same. As can be seen in the AFM images (Figure R3), phages that were treated with EtOH or PFA were indeed less twisty. We have replaced Figure 3d-e with the new one and added the phage loading information in the caption in the revised manuscript.

Figure R3. AFM images of (a) untreated M13, (b) EtOH-treated M13 and (c) PFA-treated M13. The loading amount of the three types of M13 phages was adjusted to be the same (10^9 pfu). (Added to revised manuscript as part of Figure 3c-e)

Q7. Also in Fig. 3, the simulations are not rigorously discussed in the manuscript, and they are not matched to any relevant experimental measurements. My suggestion is to remove them since they are not given proper attention in the work and the assumptions made to create the models are not transparently discussed in the main text. These data also seem redundant with the data in Fig. 3a, b.

Answer: Thank the reviewer for this inspiring comment. The simulations in Figure 3f-3h are complementary to experimental measurements in Figure 3a-b, and thereby are used to visualize the flexibility difference between the untreated M13 phage and EtOH/PFA-treated phages. While the data presented in Figure 3a, b (Stiffness and E value measured in the dry state by AFM) only shows how soft and elastic the material is, a material requires both an elongated shape and a low stiffness/ E value to show high flexibility. Therefore, simulation results, reflecting the mechanical properties in the solution state (and thus more related to the application environment for phage), are not redundant with the data in Figure 3a, b. Although AFM images can also indicate the flexibility difference between these phages, they only show the morphology difference of phage at the static state. On the contrary, the numerical simulation analysis visualized

the deformation difference between these phages under a certain external force, which was encountered when either A-f-M13-MB or A-r-M13-MB captures CTCs under mild shaking. From the simulation results, we concluded that under the same CTC incubation conditions, the untreated M13 phage has a greater spatial configurational freedom than their rigid counterparts. This allows the untreated phages to twist more when binding with CTCs and resulted in improved multivalent interaction. In summary, the simulation data (Fig. 3f-3h) present the mechanical properties of the phages from different angles compared to other data (wet state for simulation data and dry state for other data such as Figure 3a-e). So we have decided to keep the data. We have also added necessary descriptions and discussions (in the Section “Mechanical properties and dynamic movement of the engineered M13 phage”, Page 6) to clarify the simulation results in the revised manuscript.

Q8. It is nice to think that the modification of the phages with many aptamers should increase the valency of the interaction. However, there are no measurements in the manuscript directly demonstrating multivalency. This claim should be removed or direct evidence provided that multiple aptamers are able to bind to each single cancer cell thanks to the added flexibility. Given the chemical effect that PFA may have on the nucleotides and phage proteins, the "rigid" system is not a good reference control for a claim of mono- vs -multivalency. For example, there could be crosslinking happening in the PFA-treated phages that prevents the same extent of interaction as the untreated phages, regardless of rigidity.

Answer: We appreciate the reviewer for the constructive comment. **We need to clarify at first that, the "rigid" phage system is not used as the mono-valent target-binding reference control. Instead, we used a free aptamer as the mono-valent target-binding reference.** We also need to clarify that phage is a nanofiber (~7 nm wide), much smaller than the cell size (~20,000 nm), so many phage nanofibers bind to one single cell with each nanofiber providing multiple binding sites (Figure R4). According to the definition of multivalent interaction, which refers to the simultaneous binding of multiple ligands on one biological entity (a molecule or a surface) to multiple

receptors on another, the interaction between a cell and A-f-M13-MB or A-r-M13-MB is regarded as multivalent interactions. Compared with A-r-M13-MB, A-f-M13-MB presents a flexibility-enhanced multivalent interaction with cells (Figure 4a). We tried to capture the direct evidence for the flexibility-enhanced multivalent binding between a A-f-M13-loaded substrate and a single target cancer cell by AFM. As indicated by arrows in Figure R4, M13 nanofibers are twisted to adapt the shape of the cell surface for better contact and tighter binding. This observation clearly demonstrated our claim of the flexibility enhanced multivalent binding.

Figure R4. AFM image of single MCF-7 cell bound to A-f-M13-loaded interface (mica, for the ease of AFM image collection). The white arrows indicated that M13 nanofibers were twisted to adapt to the shape of the cell surface for better contact and tighter binding. Cells were cultured to adhere to the substrate for better observing the AFM images.

Q9. On page 11, the manuscript reads: "the numerous aptamers on M13 nanofibers have the capacity to replenish any degraded aptamers nearby, and they can dynamically adjust their positions by freely twisting the M13 scaffold". This is a nice concept; however, there is no rigorous experimental evidence of any of these claims in the manuscript.

Answer: We appreciate the reviewer for the constructive comment. We rephrased our statements as follows since they are not supported by direct experiments: "It is possible that the numerous aptamers on M13 nanofibers have the capacity to replenish

any degraded aptamers nearby, and they might dynamically adjust their positions by freely twisting the M13 scaffold.”

Reviewer 2:

General Comments:

In this work by Hui-Da Li et al, M13 phage is genetically modified to be used for isolation and count of CTCs from the blood from breast cancer patients, in the format of nanofibers. This can be used for efficient diagnosis of breast cancer subtypes.

Reviewer 1 specific comment:

Q1. Even though this work has novelty and is well executed, the overall idea and the field around specifically M13 phage to be used as biomaterial or relevant for diagnostics or other applications is not new. This recent review is just an example capturing the progress in the field (<https://doi.org/10.1016/j.mtbio.2023.100612>).

Answer: We appreciate the reviewer for the constructive comment and have cited the reference indicated by the reviewer (in Introduction Section, Page 3). The overall idea of this study is not like most of the M13 phage related literatures, which constructed novel M13 phage-based biomaterial for the application of diagnosis, therapy or bio-detection. **The overall aim of this paper was to solve the “never-both” dilemma faced by most of the affinity-based bioassays in mechanical point of view.**

For most of the affinity-based bioassays, including CTC isolation assay, an ideal affinity surface should simultaneously have efficient capture ability towards target and low-nonspecific adsorption due to the interferences, which requires affinity ligands and anti-fouling components to be present simultaneously. However, the simultaneous presence of the two components unavoidably leads to mutual interference, potentially compromising target-binding affinity and anti-fouling capabilities. This is the “never-both” dilemma faced by most of the affinity interfaces. **We found that flexible M13 phages energetically enhance the target cell binding affinity and entropically discouraged non-target cells adsorption.** As a consequence, the need for additional coatings of anti-fouling polymers was obviated, which saved the surface area for the

decoration of binding ligand on the interfaces at the largest extent. This “never-both” dilemma is therefore solved.

Our work is by far the first report to study phages in the viewpoint of mechanical properties, and provided a novel physical shielding strategy that was not reported before. The application scope of the affinity interfaces proposed in our study was not only limited to CTC-based cancer diagnosis but also others such as hemopurification, pathogen capture, protein isolation, etc. Since increasing evidence has shown the importance of biomechanical cues on the immune response and immunotherapy, the current work presented in this manuscript is also expected to provide inspiration to researchers in the related field.

Q2. The experiments are very carefully executed overall, the data are sound, but it is not clear how stable this genetically modified phage is over time. Is stable enough to be able to be used as a diagnostic regularly or is this a one-off study with modified phage freshly prepared for each experiment? Usually modified phages are not very stable.

Answer: We appreciate the reviewer for the constructive suggestion. **The genetically modified phage is stable over time.** The genetic modification of M13 phage was achieved by directly inserting foreign gene fragment into the phage vector M13KE. Afterwards, by infecting the phage host with the recombinant M13KE, recombinant M13 phage was obtained, which can be stored stably at -20 °C and propagated as wild type M13 phage does.

Besides genetic modification, **M13 phage was also chemically conjugated with CTC-binding aptamers** through a two-step reaction. This dual-modified phage can be stable in buffer solution at 4 °C for one week. For further evaluating its potential as a regular diagnostic tool, A-f-M13-MB were freeze-dried and preserved at -20 °C, and the loading of aptamers were monitored at different time points. As shown in Figure R5, the lyophilized A-f-M13-MB can be stable without obvious decrement in aptamer-

loading for ~7-week-preservation. Furthermore, the CTC capture efficiency (at 100 CTC/mL) maintains over 83.67% even when the aptamer-loading reduces to 520/phage. **This demonstrated the potential of A-f-M13-MB as a regular diagnostic tool, whose lyophilized form can be stable and active for ~7 weeks.** We have added the related discussion along with Figure R5 (as Supplementary Figure 4) in the revised manuscript.

Figure R5. The amount of aptamer on each A-f-M13-MB after storage for 0-48 days. (Added to revised manuscript as part of Supplementary Figure 4)

Q3. Also, how was the specific modification selected-what were the specific criteria that these sequences were selected for this phage modification-please clarify on the methods and the text.

Answer: We appreciate the reviewer for the thoughtful comment. M13 phages were modified both at the tip and at the sidewall. For the tip modification (pIII capsid protein), 6His tag, a short tag frequently adopted for protein purification were displayed for the ease of immobilizing M13 phages on Ni-functionalized solid surface in an end-on way. **The reason for choosing 6His tag is the high chelating efficiency** between 6His and Ni ion (over 85% immobilization efficiency in our case), **the moderate stability** of this chelation (can tolerate whole blood matrices without dissociation) and **the ease in operation** (do not need extra enzymes or specific reaction condition) (Reference are listed at the end of this answer).

The modification of the sidewall (pVIII capsid protein) was to endow M13 phages with breast cancer (BC) CTC capture ability. Mucin 1 (MUC1) is excessively

overexpressed in over 90% of breast tumors, thus was chosen as a biomarker for BC CTCs capture in this study. The aptamer that specifically binds to the variable number tandem repeat (VNTR) region of MUC1 was conjugated on the N-terminus of the pVIII capsid protein. **We chose the aptamer that targeting at the VNTR region of MUC1 instead of the whole MUC1 protein out of the following consideration:** The extracellular domain of MUC1 extends around 200-500 nm beyond the cell surface, with 20-120 VNTR repeats located in the N-terminal domain. The 880-nm M13 nanofibers, with repeated VNTR-targeting aptamers conjugated on the sidewall, exhibit a favorable geometric fit with MUC1. **This allowed hierarchical match between CTC cells and A-f-M13-MB at three dimensions: the molecular recognition** between VNTR repeats and their corresponding aptamers, **the sub-micron-scale fit** between MUC1 and M13 nanofibers, and **the micron-scale match** between CTCs and M13-anchored topological interfaces. As a consequence, the A-f-M13-MB has a high CTC binding affinity and CTC capture efficiency that fulfill the requirement for BC CTC isolation in clinic.

We have actually clarified the specific criteria for aptamer selection in the following two Sections: “Phage engineering and its anchoring on Ni-IDA-grafted solid surface” and “Flexible M13 facilitates better CTC capture”. Nevertheless, we have revised the related description in the text to make it more clear and described the reason for 6His tag modification in the SI file (see Section 1.3.2: Genetic modification of M13 phage).

References:

- [1] Alarcó n-Correa, M. et al. Self-assembled phage-based colloids for high localized enzymatic activity. *ACS Nano* **13**, 5810–5815 (2019).
- [2] Shen, W., Zhong, H., Neff, D. & L. Norton, M. NTA directed protein nanopatterning on DNA origami nanoconstructs. *J. Am. Chem. Soc.* **131**, 6660–6661 (2009).
- [3] Conti, M., Falini, G & Samori, B. How strong is the coordination bond between a histidine tag and Ni-nitrilotriacetate? An experiment of mechanochemistry on single molecules. *Angew. Chem. Int. Ed.* **112**, 221-224 (2000).

Q4. I understand why M13 was selected as an example phage, but are there any other potential phages that could work equally well with this methodology?

Answer: Yes, as long as the phages are **filamentous** and **flexible**, they **might have the potential** to work equally well as M13 phage do. For **spherical phages** such as T7, Q β or MS2, although aptamers might also be chemically conjugated on the capsid protein of these phages, **their affinity towards CTC may not be as high as M13 phage** because of the limited contact between CTC and the spherical phage particles. Tobacco mosaic virus (TMV), which possesses **a rod-like structure but are not flexible, cannot work equally well with M13 phages**, as TMV cannot be bent and twisted freely to have more contacts with CTC cells. **Fd phage has filamentous and flexible structure similar to M13 phages**, except for more pVIII protein (2700 pVIII protein for M13 phage, and ~4000 pVIII protein for Fd phage). Therefore, after decoration with aptamers, Fd phages **can potentially act as scaffold for CTC capture**, with flexibility-enhanced CTC binding affinity and entropically-disfavored WBC adsorption. **Nevertheless**, as longer phage length may result in larger bending upon external flow, **Fd phage might not work equally well, as it has higher entropy loss**. Actually, we are working on another project to explore the effect of phage length on CTC capture performance, and found out that the phage length does affect the CTC capture kinetics and thermodynamics. Trimmed M13 phage with appropriate length has a better CTC capture performance. Further studies are in process to elucidate this observation.

Q5. It is also a bit confusing what is the overall aim of the paper. Is it to provide a new diagnostic tool for cancer or is it to provide a new methodology that relates to M13 mechanical properties? It has to be phrased in a way that is clear and potentially more clinical samples can be tested-also from other cancers and not only breast cancer.

The methodology otherwise is sound and the methods are well written and reproducible.

Answer: We appreciate the reviewer for the constructive comment.

The current study aims at constructing an efficient and antifouling CTC isolation interfaces for the precise diagnosis and typing of breast cancer, and solving the “never-both” dilemma for elevating both the binding affinity and antifouling capacity. We have clearly stated this aim in our manuscript. As the flexibility of M13 phage endows the phage-based CTC isolation interface with high CTC binding affinity and high antifouling ability, this interface outperforms its commercial counterparts (CellSearch[®] and SE iFISH[®]) in terms of isolation capability, which reduces the misdetection rate and false negatives. The phenotyping of the released CTCs further provides a reliable way for the typing of breast cancer. **The primary results involving a clinical cohort of 100 participants in this study demonstrated the diagnosis potential of this CTC isolation interfaces, while further study involving multicenter trial with wider participants is now in-process to validate whether it can be used as a real diagnostic tool.**

Q6. Some further comments: figure 6 (e) is not very clear-please demonstrate the result clearly. figure 7 (d) some text on the side is cut-please amend.
supplementary figure 4 (a-c) (d-f)-f is missing from parenthesis, please amend and increase contrast so that is obvious what you want to demonstrate.

Answer: We appreciate the reviewer for the thoughtful suggestion. We have conducted the cell viability test again, got better fluorescence microscope images and presented them in a clearer way (See Figure R6). We have also amended the mistakes of figure 7(d) (the cut texts) and supplementary figure 4 (Renamed as supplementary figure 5 in the revised manuscript, the missing f), adjusted the contrast of supplementary figure 4 as the reviewer suggested. We have thoroughly checked all figures in the main text and the supporting information file to prevent similar errors (missing letters and cut text).

Figure R6. Representative fluorescence microscope images of the isolated live cells and dead cells. The live cells emitted green fluorescence and dead cells (indicated by arrows) emitted red fluorescence. Scale bar: 100 μm. The dead cell was further enlarged as in the white square frame. Insert is the cell viability of the isolated MCF-7 cells calculated from PI/AO staining results. (Added to revised manuscript as part of Figure 6d-e)

Reviewer 3:

General Comments:

In this manuscript, a versatile phage nanofiber system was utilized to enhance the capture efficiency of CTCs and inhibit the non-specific adsorption of WBCs by modifying the high peak degree of aptamer on the nanostructures of this fiber to increase the binding affinity between the ligands on the functional microspheres and the markers of the circulating tumor cell membranes with the help of the nanofiber's flexibility and deformability. **This approach is very interesting and the newly-developed probes exhibit an overwhelming strength in CTCs capture over the previously reported nanoworms-like probes due to much more flexible chain of phage M13. The authors provide adequate experimental data to testify the feasibility of the probes in highly efficient enrichment of CTCs based on the samples with spiked tumor cells and clinical patient samples.** However, to be one of the valuable articles in the research field of CTCs capture, the following key issues should be considered in order to attract more readers, because some microfluidic microarray-based systems have already been able to realize the capture of high-purity

and high-sensitivity CTCs for prognostic monitoring applications. Additionally, as for rapid culture and proliferation of CTCs in vitro after isolation from clinical patients' blood samples for precise diagnosis and early warning related to metastatic recurrence of cancer patients, or the use of proliferated CTCs cell spheres to construct a mouse transplantation xenograft to study new intervention methods of tumor metastasis, all of these require that the release and detachment of CTCs from isolation probes can be achieved in a highly active and efficient manner. Although with some advantages in improving purity and capture efficiency of CTCs compared to ordinary magnetic probes, this probe still needs to be improved in terms of the downstream applications requirements. So, this manuscript can't be accepted for publication in this journal before following concerns and questions have been well addressed.

Response:

We thank the reviewer for the positive comments. **We agree with the reviewer that efficient release of the captured CTC with high viability and activity is crucial for downstream applications including in vitro culture of CTCs or mouse transplantation xenograft construction. Therefore, we thoroughly evaluated the viability, migration ability and the phenotype drifting profile after the CTCs were released and re-cultured. The results indicated that our CTC isolation system meet the downstream applications requirements** (With detail shown in the response for Q2).

Responses to the reviewer's specific comments are as follows:

Reviewer 1 specific comment:

Q1. The quality of the figures needs to be carefully checked and improved, such as several panel numbers are missing in Figure 5; the curves in panel e are not labeled, and the vertical axes are not labeled with physical quantities. Similar errors happened to Figure 3a, Figure 6 and 7.

Answer: We appreciate the reviewer for the thoughtful suggestion. We have thoroughly checked all figures in the main text and the supporting information file to

prevent similar errors. As for Figure 5, panel e, RDF is unitless and thus the vertical axes were not labeled with physical quantities. We have corrected all of other errors pointed out by the reviewer.

Q2. As a powerful tool of CTCs enrichment, it is also necessary to consider to carry out the high availability release, the special structure and reactivity of M13 phage as well as the flexible characteristics contributed to higher binding affinity of CTCs, how to realize such CTCs release from the probe after CTCs enrichment upon the M13 phage-based system?

Answer: We agree with the reviewer that the high availability release of the captured CTC is crucial for a powerful CTCs enrichment system. An ideal release method should be mild enough to maintain the CTC with high viability, the ability for in vitro proliferation and migration. Therefore, **we chose deoxyribonuclease I (DNase I) to mildly release CTC from the probe after CTC enrichment.** DNase I can nonspecifically digest ds/ss DNA by hydrolyzing phosphodiester bond. In our study, M13 phages were decorated with CTC-binding aptamer for CTC capture. **Upon the addition of DNase I, aptamers were hydrolyzed and the captured CTC was thus released.** Because **DNase I hydrolyzes aptamers instead of degrading cells, the released CTCs maintain high viability**, which was demonstrated by the viability test and in vitro cultivation of the released CTCs (Figure 6d-f). We also evaluated the migration ability and the phenotype drifting profile after the CTCs were released and re-cultured. As shown in Figure R7a-c, the expression level of estrogen receptor protein (ER, feature biomarker of MCF-7 cell) showed no difference after CTC isolation and re-culture, indicating the phenotype was stable and not drifted after treatment with A-f-M13-MB. Namely, the isolated CTCs can represent the feature of original ones. From the analysis results indicated in Figure R7d-f, it is clear that the CTCs after treatment with A-f-M13-MB have comparable or even better migration ability compared to the original CTCs. The high viability of the isolated CTCs can well meet the requirement for downstream applications including in vitro CTCs culturing or mouse transplantation xenograft construction. We have added the related results and discussion in Section

Figure R7. The phenotype drifting profile and comparison in migration ability of CTCs before and after isolation by A-f-M13-MBs and re-culture. The fluorescence images (a) and relative fluorescence intensity (b) of MCF-7 cells before (labeled as “control”) and after isolation by A-f-M13-MBs and being re-cultured for 12 days (labeled as “Capture-release”). Cells were stained by DAPI and anti-ER antibody. The two groups showed no significance, indicating identical ER expression level. Scale bar: 50 μm . (c) Western blotting analysis of ER in MCF-7 cells before (labeled as “control”) and after isolation by A-f-M13-MBs and being re-cultured for 12 days (labeled as “Capture-release”). GAPDH was used as an internal reference. (d) Quantitative analysis of the expression level of ER based on the relative gray level of ER band with respect to that of GAPDH in western blotting results. The two groups showed no significance, indicating identical ER expression level. (e) The migration pathway images of MCF-7 cells before (labeled as “control”) and after isolation by A-f-M13-MBs and being re-cultured for 12 days (labeled as “Capture-release”). Quantitative analysis of the mean distance (f) and mean speed (g) of cells. **** $p < 0.0001$. The mean distance of the two groups of cells showed no significance, but the migration speed of the cells after isolation and re-culture was a little faster than the control cell, indicating higher viability.

Q3. It is recommended that the curve in Figure 7c be replaced with an AUC curve and the corresponding table of raw data be provided.

Answer: We presume the reviewer means ROC curve when mentioning AUC curve, which is exactly was we present in Figure 7c. Briefly, BC patients were set as the positive outcome (1), whereas both healthy donor and begin patients were set as the negative outcome (0). By setting the cutoff value of CTC number $> 4/\text{mL}$, the diagnosis results of our method can be obtained (See Table R1). By comparing our diagnosis results with that by the hospital, true positive ratio (sensitivity) and true negative ratio (specificity) can thus be calculated, and the ROC curve can thus be obtained. We processed the raw data with MedCalc software. The ROC curve was revised due to the correction of one diagnosis results, and the corresponding raw data were also provided in the revised SI file as the reviewer suggested

Table R1 The raw data of AUC curve.

Sample ID	Clinical diagnosis results	Diagnosis results by our method	Number of CTCs/mL (Cut off >4)
1	1	1	16
2	1	1	5
3	1	1	21
4	0	0	2
5	1	0	4
6	1	1	14
7	1	1	12
8	0	0	1
9	1	1	15
10	1	1	7
11	1	1	22
12	1	1	24
13	1	1	31
14	1	1	35
15	1	1	41
16	1	1	47
17	0	1	5
18	0	0	3
19	1	1	29
20	1	1	6
21	1	0	4
22	1	1	24
23	0	0	3
24	1	1	5
25	1	1	7

26	0	0	1
27	0	0	2
28	1	1	6
29	1	1	20
30	0	0	1
31	0	0	1
32	0	0	2
33	1	1	8
34	0	0	4
35	0	0	2
36	0	0	0
37	1	0	3
38	0	0	1
39	1	0	4
40	1	1	55
41	1	1	19
42	1	1	16
43	0	0	1
44	0	0	4
45	0	0	0
46	1	1	11
47	1	1	6
48	1	1	8
49	1	1	14
50	0	0	3
51	1	1	13
52	1	1	12
53	1	1	6
54	0	0	0
55	0	0	0
56	0	1	5
57	1	1	6
58	0	0	0
59	1	1	5
60	0	0	1
61	1	1	21
62	1	1	13
63	0	0	1
64	0	0	0
65	1	1	28
66	0	0	0
67	0	0	3
68	1	1	43
69	1	1	35

70	1	1	7
71	1	1	15
72	1	1	42
73	1	1	23
74	1	1	15
75	1	1	47
76	0	0	1
77	0	0	0
78	0	0	2
79	1	1	19
80	1	1	18
81	0	0	3
82	0	0	0
83	0	0	4
84	1	1	13
85	1	1	32
86	1	1	10
87	0	0	3
88	1	1	59
89	1	1	19
90	1	1	23
91	0	0	0
92	0	0	0
93	0	0	0
94	0	0	0
95	0	0	0
96	0	0	0
97	0	0	0
98	0	0	0
99	0	0	0
100	0	0	0

Note: BC patients were set as the positive outcome (1), whereas both healthy donor and benign patients were set as the negative outcome (0).

Q4. The authors claimed that “An average of approximately 622 aptamers was found on the sidewall of each Apt-M13 (Supplementary Fig. 3), how did they quantify the M13?”

Answer: We apologize for not clearly describing the calculation method of the aptamer loading number per phage. **The number of M13 phages was quantified by titer.** For a typical calculation of the number of aptamers decorated on each M13 phage,

5×10^{10} pfu (determined by titrating) of N₃-M13 was first anchored on 1 mL 2.5mg/mL of MB suspension. After the anchoring step, we collected the supernatant to get the number of M13 phages that were not anchored on MBs by titer. The number of M13 phages anchored on MBs can therefore be obtained by subtraction. Meanwhile, we estimated the amount of MBs in 100 μ L of suspension to be 2.567×10^5 by microscope observation. Therefore, we can know how many M13 phages are anchored on each MB (~16714 phages for each MB).

Afterwards, the N₃-M13-MBs further reacted with excessive FAM-labeled DBCO-aptamer (FAM-Apt-DBCO). We measured the fluorescence intensity of the supernatant after reaction to obtain the amount of the unreacted FAM-Apt-DBCO, so the amount of the loaded aptamers on N₃-M13-MBs can be obtained by subtraction. In this way, the amount of aptamers on each M13 phage can be readily obtained ($1.04 \times 10^7 / 16714 = 622$). We have revised the related descriptions regarding quantification method of the aptamers on each phage (Section S2. Aptamer loading amount on M13 with different stiffnesses in the SI file) to make this manuscript more readable.

Q5. As we know, CTCs have different EMT subtypes, and a large number of aptamer molecules are able to be attached to this M13 phase nanofibers, it is strongly suggested to examine whether it is possible to isolate CTC subphenotypes of cells covering different EMT characteristics from the clinical patients' blood samples if several ligands targeting E-type and M-type CTCs were attached to a single phage chain at the same time?

Answer: We appreciate the reviewer for the constructive suggestion. The proposed CTC isolation system targeting MUC1 has the potential to isolate CTC subphenotypes covering different EMT characteristics, as was evidenced by the isolation and identification of EpCAM-negative CTCs from three patients (Patient No. 16, No. 72 and No. 88) that suffer from severe lymphatic metastasis (Figure S19).

In order to better evaluate the ability of the current system to isolate CTCs with different EMT subphenotypes, E-type CTC model cell (MCF-7 cell, EpCAM⁺, N-

cadherin⁻) and M-type CTC model cell (MDA-MB-231 cell, EpCAM⁻, N-cadherin⁺) were mixed with different ratios and spiked in CTC-free whole blood, followed by the isolation with A-f-M13-MB and immunofluorescence staining. As shown in Figure R8a, the CTC mixtures maintained high capture efficiency regardless of the ratio of E-type/M-type, and the subphenotypes can be clearly identified by immunofluorescence images (Figure R8b). This clearly demonstrated that the current CTC isolation system targeting MUC1 can isolate CTC subphenotypes covering different EMT characteristics.

We also constructed A-f-M13-MB with aptamers targeting E-type and M-type CTCs attached to a single phage chain at the same time, as the reviewer suggested. To be more specific, the anti-vimentine aptamer (targeting M-type CTC) and anti-EpCAM aptamer (targeting E-type CTC) were first assembled to form a Y-shaped DNA scaffold (Figure R9a, b), and then attached to M13 phages via the same procedure used in the current study. The reason for assembling two aptamers into one Y-shaped DNA assembly is to keep the molar ratio of the two aptamers to be 1:1. The resultant A-f-M13-MB has similar isolation performance toward CTC mixtures (Figure R9c), and is capable of identifying the subphenotypes through immunostaining (Figure R9d).

It is worth noting that by incorporating two specific cutting sites into the sequence of Y₁ and Y₂ (at the position indicated in Figure R10), E-type and M-type CTCs might be released separately by the corresponding restriction endonucleases. Nevertheless, considering that the EMT or MET process is a consecutive process, i.e., the EMT subtypes of CTCs in the blood of cancer patients might not just simply be classified as E-type, M-type and the intermediate-type. Therefore, “capture them all and then verify their phenotype by immunostaining” are more suitable to fulfill the reviewer’s requirement. We have added the experimental results regarding the capture and verification of E-type/M-type CTC mixtures by A-f-M13-MB in the revised manuscript as a new section “Isolation of CTCs with different EMT subphenotypes by M13-MBs” on page 14.

Figure R8. The capture and identification of E-type/M-type CTC mixtures by A-f-M13-MB. (a) The capture efficiency of E-type CTC and M-type CTC by capturing the CTC mixtures by A-f-M13-MB. The ratio of E-type: M-type=9:1, 3:1, 1:1, 1:3 and 1:9. (b) Representative fluorescence microscope image of the isolated CTC mixture. The ratio of E-type: M-type=1:1, the E-type CTCs were immunostained by Anti-E Cadherin antibody to emit green fluorescence, whereas the M-type CTCs were labeled with red fluorescence by Anti-N Cadherin antibody. Scale bar: 50 μ m.

Figure R9. The capture and identification of E-type/M-type CTC mixtures by A-f-M13-MB loaded with Y-shaped DNA assemble. (a) Illustration of the construction of Y-shaped DNA assemble. The sequence of anti-vimentine aptamer and anti-EpCAM aptamer were incorporated into scaffold DNA Y_1 and Y_2 , respectively. The three scaffold DNA Y_1 , Y_2 and Y_3 were partially hybridized with each other, thus forming the Y-shaped DNA assemble. (b) Electrophoresis analysis indicates the successful

assembly of the Y-shaped DNA assemble. Lane 1: marker; lane 2: Y₁; lane 3: Y₂; lane 4: Y₃; lane 5: Y₁₊₃; lane 6: Y₁₊₂₊₃. (c) The capture efficiency of E-type CTC and M-type CTC by capturing the CTC mixtures by A-f-M13-MB loaded with the Y-shaped DNA assemble. The ratio of E-type: M-Type=9:1, 3:1, 1:1, 1:3 and 1:9. (d) Representative fluorescence microscope image of the isolated CTC mixture. The ratio of E-type: M-Type=1:1, the E-type CTCs were immunostained by Anti-E Cadherin antibody to emit green fluorescence, whereas the M-type CTCs were labeled with red fluorescence by Anti-N Cadherin antibody. Scale bar: 50 μm. Inserted are enlarged images of E-type CTC and M-type CTC. Scale bar: 10 μm. (Added to revised manuscript as part of Supplementary Figure 20)

Figure R10. Illustration of the two specific cutting sites of Y₁ and Y₂ of Y-shaped DNA.

Table R2 All the DNA sequences used in Y-shaped DNA.

Name	DNA Sequence (5'-3')
Y ₁	CACGCATAGCCTTTGCTCCTCGTCTGGAACGTCGCAGCTT TAGTTCTGGGCCTATGCGTGT TTTTTTGTAGTCGGTACCTAA GACTTCTGAGCATGCACTGAC
Y ₂	CACTACAGAGGTTGCGTCTGTCCCACGTTGTCAT GGGGGTTGGCCTGT TTTTTTGTCAGTGCATG CTCAGAAGAACTCACGTGACG
Y ₃	DBC0-ATTGCGTATGTCACGTCACGTGAGTTCGTCTTAGGT ACCGACTAC

Underline indicated the sequence of anti-vimentine aptamer (in Y₁) and anti-EpCAM aptamer (in Y₂), respectively.

REVIEWERS' COMMENTS

Reviewer #1 (Remarks to the Author):

I appreciate the extensive answers provided by the authors in response to my comments. The changes made to the manuscript were responsive to my critiques and have improved the clarity of the experimental design. The results are rigorous and broadly support the claims made. Additionally, claims that rested on missing data have been removed in this revised version, which I enjoyed reading very much. I recommend publication as is.

Reviewer #2 (Remarks to the Author):

The authors have addressed my concerns and I am therefore happy to approve this work for publication.

Reviewer #3 (Remarks to the Author):

In this revised manuscript, all my concerns and suggestions have been addressed and involved. The authors made a detailed response to the comments in the letter as well. So I am glad to suggest to accept this manuscript.

Response to reviewers' comments:

REVIEWERS' COMMENTS

Reviewer #1 (Remarks to the Author):

I appreciate the extensive answers provided by the authors in response to my comments. The changes made to the manuscript were responsive to my critiques and have improved the clarity of the experimental design. The results are rigorous and broadly support the claims made. Additionally, claims that rested on missing data have been removed in this revised version, which I enjoyed reading very much. I recommend publication as is.

Response: Thank you for suggesting the acceptance of our manuscript.

Reviewer #2 (Remarks to the Author):

The authors have addressed my concerns and I am therefore happy to approve this work for publication.

Response: Thank you for suggesting the acceptance of our manuscript.

Reviewer #3 (Remarks to the Author):

In this revised manuscript, all my concerns and suggestions have been addressed and involved. The authors made a detailed response to the comments in the letter as well. So I am glad to suggest to accept this manuscript.

Response: Thank you for suggesting the acceptance of our manuscript.